# MARVEL: Multi-Agent RTL Vulnerability Extraction using Large Language Models

## Abstract

Hardware security verification is a challenging and time-consuming task. Design engineers may use formal verification, linting, and functional simulation tests, coupled with analysis and a deep understanding of the hardware design being inspected. Large Language Models (LLMs) have been used to assist during this task, either directly or in conjunction with existing tools. We improve the state of the art by proposing `MARVEL`, a multi-agent LLM framework for a unified approach to decision-making, tool use, and reasoning. `MARVEL` mimics the cognitive process of a designer looking for security vulnerabilities in RTL code. It consists of a supervisor agent that devises the security policy of the system-on-chips (SoCs) using its security documentation. It delegates tasks to validate the security policy to individual executor agents. Each executor agent carries out its assigned task using a particular strategy. Each executor agent may use one or more tools to identify potential security bugs in the design and send the results back to the supervisor agent for further analysis and confirmation. `MARVEL` includes executor agents that leverage formal tools, linters, simulation tests, LLM-based detection schemes, and static analysis-based checks. We test our approach on a known buggy SoC based on OpenTitan from the Hack@DATE competition. We find that of the 51 issues reported by `MARVEL`, 19 are valid security vulnerabilities, 14 are concrete warnings, and 18 are hallucinated reports.

## 1 Introduction

Detection of hardware security vulnerabilities in Register-Transfer Level (RTL) designs is a time-consuming and challenging process (Dessouky et al., 2019). Considerable research efforts have been dedicated to making this validation process easier. These include using deterministic methods like formal verification (Iyer et al., 2019; Sturton et al., 2019; Ray et al., 2019), information flow tracking (Hu et al., 2021; Brant et al., 2021), fuzzing (Muduli et al., 2020; Tyagi et al., 2022), and, more recently, non-deterministic methods involving the use of large language models (LLMs) (Ahmad et al., 2025; Fu et al., 2023; Tarek et al., 2025). LLMs have been used as debuggers/linters (Fang et al., 2025). They may use guidelines for the detection of bugs, such as Common Weakness Enumerations (CWEs), or may use external information (outside of source code), such as design specifications, to aid in detection (Tarek et al., 2024; Akyash & Kamali, 2024). They may use external tools such as linters, formal verification, and/or simulators to assist them in debugging (Xu et al., 2025). While these approaches are promising, they require a preset action plan. The LLM's decision-making process may help determine whether a segment of code is insecure, but it does not control the workflow for tool use or information gathering. This gap in autonomous thinking can be addressed by using LLMs in an agentic workflow. Giving an LLM the ability to plan, call tools, and control information flow lets it act like a human analyst and make *human-like* decisions (Wang et al., 2024). This better simulates the thought process of an RTL designer or verification engineer while debugging a digital design. Bug-hunting might involve multiple tools to iteratively localize the cause of a misbehaving design. This process can be emulated in a multi-agent framework (Li et al., 2024).

`MARVEL` is a multi-agent framework that implements a unified approach to decision-making, tool usage, and reasoning towards the goal of RTL bug detection. It uses a *Supervisor-Executor* architecture (LangGraph, 2024). The supervisor manages communication and coordination among specialized executor agents. Each executor uses a unique bug detection strategy coupled with the tools required to implement its strategy. **Linter Agent**, **Assertion Agent**, **CWE Agent**,

**Similar Bug Agent**, **Anomaly Agent**, and a **Simulator Agent** are the executor agents. The supervisor agent identifies the security objectives relevant to the design by traversing through directories, source code, and design specification documents. Then it calls one executor agent at a time to identify vulnerabilities that violate security objectives. The supervisor may use multiple executors before determining whether the security objective is satisfied. In this process, security bugs are identified, and a report is provided to the user summarizing the security issues. An overview of the multi-agent supervisor-executor flow is shown in Figure 1.

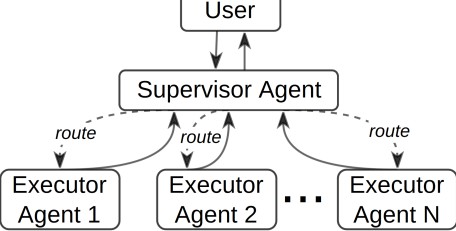

While a similar strategy has been used for automated bug repair for software (Lee et al., 2024), research efforts for hardware description languages (HDLs) do not present a comprehensive approach. Unlike software bugs, RTL bugs are deeply tied to hardware semantics such as clocking and concurrency, which make them harder to detect and repair with traditional software-centric approaches. Moreover, the consequences of RTL bugs can propagate to silicon, where fixes are expensive and time-

Figure 1: Supervisor-Executor Architecture.

consuming. We develop solutions for RTL by integrating RTL static analysis tools and handling the unique challenges faced with HDLs. These include identifying hardware security objectives, using hardware CWEs, forming security assertions, and reasoning about the outputs from a digital design perspective. MARVEL integrates with digital workflows and leverages existing infrastructure to provide RTL vulnerability detection. The key contributions of this work are:

- First multi-agent bug detection framework (MARVEL) for hardware designs (Section 2).
- Evaluation of MARVEL on the Hack@DATE 2025 OpenTitan SoC (Section 4.1).
- Architecture analysis to evaluate the benefits of each agent in MARVEL (Section 4.2).
- Open-sourcing implementation and results for the community through our repository.

## 2 MARVEL

We propose MARVEL, i.e., **M**ulti-**A**gent **R**TL **V**ulnerability **E**xtraction using **L**LMs. We implement MARVEL with a *Supervisor-Executor* architecture. This architecture decomposes the larger verification task into multiple sub-tasks and allows each executor to specialize for a given task (e.g., writing and running assertions). The supervisor orchestrates the analysis to ensure cohesion and a logical sequence of actions. The components of MARVEL are shown in Figure 2.

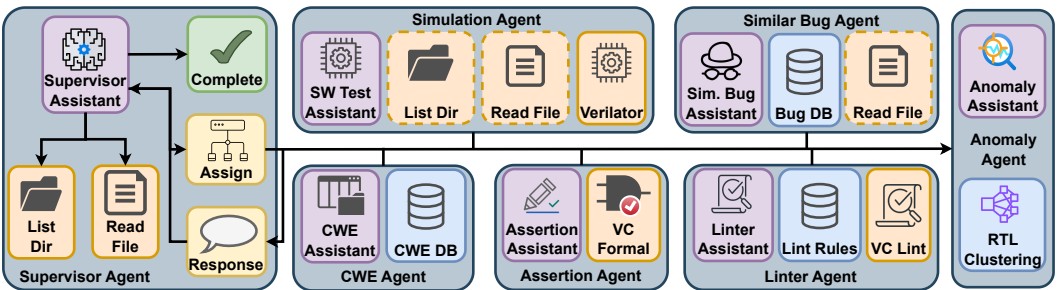

Figure 2: MARVEL's Multi-Agentic Framework. Purple denotes LLM assistants, Orange denotes tools, and Blue denotes RAG databases. The Supervisor Agent can list directories, read from files, and assign tasks to executor agents. From the responses, it may decide to continue assigning tasks or determine that the security analysis is complete. Simulator, Similar Bug, CWE, Assertion, Linter, and Anomaly agents are executor agents, each responsible for a specific security verification task.

### 2.1 OVERVIEW

LLM-based agents are typically explained using four elements: Profile, Memory, Planning, and Action. Not only does MARVEL use these modules holistically, but all seven agents of MARVEL utilize these components individually as well. Each agent consists of an assistant and tools. The assistant is an LLM agent with a specific objective, responsible for decision-making, capable of calling one or more tools. The tools are responsible for carrying out the assistant's recommended actions when called and replying with the result or with an error message.

**Profile:** describes the roles of an agent, which are usually indicated in the prompts to the LLM. This may include priming the LLM as a domain expert and providing it with its overall big picture task. For `MARVEL`, all agents (supervisor and executors) are profiled using system prompts.

**Memory:** stores information obtained from the environment and leverages the recorded memories to facilitate future actions. Short-term memory is the information present in the context window that the LLM is provided in the prompt. Long-term memory is external vector storage that agents can query and retrieve from. Retrieval Augmented Generation (RAG) is used by the respective agents to retrieve relevant information from these databases. For `MARVEL`, the external vector storages are i) CWE Database used by the CWE Agent, ii) known Bug Database used by the Similar Bug Agent, and iii) Lint Rules Database used by the Linter Agent.

**Planning:** is the decision-making process based on the information available to the LLMs in the context window. LLMs use this reasoning ability to make judgments about the workflow path to take, the tools to use, and when to conclude that a task is done. For the supervisor agent, this consists in deciding which executor agent to use, making a conclusion about a reported security issue, and deciding when it is time to produce the final report. For each executor agent, the corresponding assistant does this planning, determining whether the tool needs to be called and whether there is enough information to report back to the supervisor.

**Action:** translates the agent's decisions into specific outcomes. This can be done through using external tools or by the LLMs themselves. The tools for the supervisor agent include executor agents, and the tools for the executor agents are the linter, formal, simulator, and clustering tools.

## 2.2 SUPERVISOR AGENT

The *supervisor* orchestrates the *executor agents* to perform the security analysis. The supervisor agent's system prompt is shown in Figure 3. Beyond coordination, the supervisor agent is also responsible for identifying relevant security properties from the design documentation to create a test plan and call upon the executor agents accordingly. The supervisor can explore the hardware design by listing folder content and reading files to achieve this. Files can be retrieved with and without annotated line numbers. The latter is helpful for code files, helping the agent report buggy line numbers. The executor agents return short reports describing potential issues; the supervisor can then accept an issue, escalate the analysis by invoking additional executor agents, or inspect the code directly to refine or validate the finding. Ultimately, the supervisor agent produces a report on the security issues in the design. The system prompt hints at the order and use of executor agents, but the supervisor agent may call them in any order and any number of times.

```
Supervisor's System Prompt

You are a supervisor agent in a multi-agent system focused on identifying
hardware security vulnerabilities in RTL code.  Your objective is to analyze
the given SoC and generate a detailed security report.
You have access to the following tools:  <tool_list>.
Each tool specializes in a specific task:
<tools description> Instructions for analysis:
- Read the documentation to identify security features and register interface
policies.
- Use Verilator, Assertion, Anomaly and Linter agents to uncover initial issues.
- If a bug is detected but not localized, use the CWE Agent to further inspect
the related security aspect in the surrounding RTL.
- After detecting any bugs, use the Similar Bug Agent to scan similar files (of
the same or of different IPs) for similar vulnerabilities.
Output Format:
<output_format_instructions> When your analysis is complete, end your response
with "END".
```

Figure 3: Supervisor Agent's System Prompt. It is instructed to analyze given SoC for security bugs. It is provided information about the executor agents and is tasked to produce a security report.

## 2.3 LINTER AGENT

The *linter agent* automates lint-based security analysis by (i) identifying lint checks relevant to a given security objective identified by the supervisor agent and (ii) analyzing and filtering the warnings and errors produced by the linting process. These two capabilities enable the agent to focus the analysis on the security intent and design context, thereby reducing the high false-positive rate typically observed in lint tools. To implement these capabilities, the linter agent is composed of three components: the *linter assistant*, the *lint tags retriever*, and the *lint checker*. Given a security objective and a source code file, the linter assistant coordinates the workflow. It can invoke the lint tags retriever or the lint checker in any sequence, up to a maximum of six

iterations. The lint tags retriever maps the security objective to a set of relevant lint checks by searching an indexed description of available lint rules and selecting at most 20 tags per query. The linter assistant then uses the retrieved tags to call the lint checker, which runs lint analysis on the provided code and returns any violations. If violations are found, the linter assistant performs an additional reasoning step to filter out false positives. We instantiate this architecture using Synopsys's VC SpyGlass Lint tool (Lint, 2022). The lint tags retriever operates over a database of 1255 SpyGlass lint rules (each represented by an identifier and a short description), and the lint checker executes SpyGlass with the selected tags and target module. Any errors produced by the tool, such as unknown tag names or incorrect module specifications, are returned to the linter assistant for debugging. An example of the linter agent's operation on the ADC Control FSM module is shown in Section A.1.1.

## 2.4 ASSERTION AGENT

The *assertion agent* automates assertion-based security analysis by (i) generating meaningful SystemVerilog assertions tailored to a given security objective and (ii) checking these assertions against the RTL to identify security violations. This enables targeted formal analysis that directly connects the semantics of the security objective to observable design behavior. To implement these capabilities, the assertion agent consists of two components: the *assertion assistant* and the *assertion checker*. Given a security objective and a source code file, the assertion assistant creates relevant SystemVerilog concurrent assertions based on a canonical assertion structure and the semantics of the security objective. The assertion agent may call the assertion checker repeatedly, up to six iterations, until a falsified assertion is produced or no further progress can be made. After each check, the assertion assistant inspects the results to determine whether the generated assertions uncovered a security issue, whether new assertions should be formed for deeper refinement, or whether an error in assertion syntax or binding requires correction. We instantiate this architecture using Synopsys' Formal Property Verification (FPV) tool (Formal, 2024). The assertion checker binds the generated assertions to the RTL, gathers the necessary design dependencies, populates a Tcl template with the appropriate top module, clock, and reset signals, and executes FPV. The results file, containing any falsified assertions, is then returned to the assertion assistant. If the tool encounters issues—such as syntactically invalid assertions or incorrect bindings—an exception message is sent back for debugging. An example flow of the assertion agent for the `hmac_reg_top` module is shown in Section A. 1.2.

## 2.5 CWE AGENT

The *CWE agent* supports vulnerability classification by (i) identifying the CWE most relevant to a given security objective and RTL module, and (ii) retrieving detailed descriptions and examples to contextualize potential weaknesses. This allows the agent to map design-level issues to standardized hardware-relevant CWEs, providing consistent terminology for reporting and remediation. To implement these capabilities, the CWE agent consists of two components: the *CWE assistant* and the *CWE details retriever*. Given a security objective and a source code file, the CWE assistant coordinates the workflow and may invoke the details retriever up to six times. The assistant first determines which CWE category is most relevant to the suspected security issue and then augments this classification by assessing whether the RTL exhibits behaviors consistent with that CWE. The details retriever performs two main operations: it identifies a candidate CWE-ID based on the security objective, and it then retrieves the corresponding extended description, examples, and repair patterns. We instantiate this architecture using information from MITRE's CWE database. We construct a text file in which each segment contains a CWE identifier and its description. This file is chunked using a recursive character splitter into segments of size 50, with no overlap, and with a custom separator that delineates individual CWEs. These chunks are stored in a vector database, allowing the retriever to identify the most relevant CWE based on embedding similarity, selecting a single best match. Once identified, the retriever augments the CWE with its extended description and examples, and this enriched output is returned to the CWE assistant. An example of a CWE agent run is shown in Section A.1.3.

## 2.6 SIMILAR BUG AGENT

The *similar bug agent* automates the detection of recurring bug patterns by (i) identifying RTL lines semantically similar to a known buggy line and (ii) evaluating whether these analogous lines also constitute bugs. This enables pattern-based vulnerability discovery, where issues identified once can be efficiently propagated across the design. To implement these capabilities, the agent consists of the *similar bug assistant* and the *similar bug tool*. Given a buggy reference line and

a target RTL file, the assistant coordinates the analysis and may repeatedly invoke the similar bug tool until the search completes or additional context is required. The assistant first requests candidate lines that are semantically similar to the known bug and then determines whether these candidates reflect the same underlying issue. When ambiguous, the assistant may inspect the surrounding code to refine its judgment before producing a final list of confirmed buggy locations. We instantiate this architecture using an embedding-based semantic search approach. The similar bug tool reads the RTL source file, splits it into individual lines, and embeds each line using OpenAI embeddings. These embeddings are stored in an in-memory vector store, over which a retriever returns the top ten lines most similar to the input bug based on embedding similarity. The tool annotates each matched line with its line number and returns them to the assistant. If no similar lines are found or if the tool encounters an error (e.g., missing file), a corresponding message is sent back. Otherwise, the assistant inspects the similar lines and produces a report for the supervisor agent. An example of the similar bug agent's operation is shown in Section A.1.4.

### 2.7 ANOMALY AGENT

The *anomaly detection agent* detects unexpected or atypical RTL constructs by (i) grouping semantically similar lines of code and (ii) identifying outliers that may indicate potential security vulnerabilities. This enables pattern-independent detection of suspicious behavior that may not be captured by lint rules, assertions, or known bug patterns. To implement these capabilities, the agent consists of the *anomaly detector assistant* and the *RTL clustering tool*. Given a security objective and an RTL source file, the assistant orchestrates the analysis and may repeatedly invoke the clustering tool as needed. The assistant examines anomalous constructs returned by the tool and determines whether they plausibly represent security-relevant issues, taking into account the surrounding design context when necessary. We instantiate this architecture using an embedding-based clustering approach. The RTL clustering tool first extracts all `assign` statements from the RTL, then generates embeddings for each using OpenAI's `text-embedding-3-small` model. These embeddings are clustered using DBSCAN with cosine similarity as the distance metric, grouping semantically related constructs and flagging outliers as anomalies. The tool returns both the anomalous lines and the clusters they were compared against. If no anomalies are found or if an error occurs (e.g., missing file), a corresponding message is sent back to the assistant. Otherwise, the assistant analyzes the anomaly data to determine whether any of the outliers represent potential security vulnerabilities and produces its final assessment. An example flow of the anomaly agent on the HMAC Register Top module is shown in Section A.1.5.

### 2.8 SIMULATOR AGENT

The *simulator agent* identifies potential security issues in RTL through dynamic analysis by (i) selecting and executing relevant simulation tests and (ii) interpreting failing behaviors to determine whether they correspond to security-related bugs. This enables the agent to uncover issues that manifest only under specific execution conditions and may not be caught by static analysis. To support these capabilities, the simulator agent comprises the *simulator assistant* and the *Verilator tool*. Given the name of a target IP block, the assistant coordinates the simulation workflow and may call the Verilator tool iteratively until no further analysis is required. The assistant examines failing test outputs and determines whether the observed behaviors represent security vulnerabilities, optionally requesting additional simulation runs if the diagnostics are inconclusive. We instantiate this architecture using Verilator as the simulation backend. The Verilator tool operates in two stages: it first retrieves all available software tests associated with the target IP by issuing a filtered `bazel query`. Then it runs these tests under Verilator using a `bazel test` command. The tool returns the simulation output, including logs for any failing tests. If no tests are found or the simulation run fails, a message is sent to the assistant. Otherwise, the assistant analyzes the failing behaviors to determine whether they reflect security-relevant issues and provides a final summary of identified concerns, including explanations and references to the affected RTL. An example of a simulator agent run is shown in Section A.1.6.

## 3 EXPERIMENTAL SETUP

### 3.1 BENCHMARK

We evaluate MARVEL on a vulnerable OpenTitan *earlgrey* System-on-Chip (SoC) design (lowRISC contributors, 2023) obtained from the finals of the Hack@DATE 2025 competition. Hack@DATE is a premier hardware security capture-the-flag competition. They provide contestants with an SoC design with manually inserted vulnerabilities akin to those found in real, deployed products. The earlgrey SoC is a high-quality, open-source Root-of-Trust design that provides robust hardware

security features. It utilizes the *Ibex* RISC-V processor as its main core. It integrates intellectual property (IP) block peripherals, including crypto accelerators for AES, system management units for clock, power, and reset, and I/O protocols such as SPI. The SoC has an array of security features, including end-to-end data integrity, secure boot, and first-order masking of side-channels.

The IPs we analyze with MARVEL are summarized in Table 1. We report the total number of files, design files, design LoC and number of bugs for each IP. Design files only include those used to implement the IP (i.e., excludes test files), and design LoC is the line count in those files, excluding comments and whitespace. We select these 12 IPs because they represent a wide range of functionality, spanning cryptography (e.g., AES), I/O (e.g., ADC), and system management (e.g., lifecycle controller). Bugs are spread unevenly across the selected IPs, which allows us to evaluate the effectiveness of MARVEL on IPs with zero, a few and up to ten bugs.

Table 1: IPs from OpenTitan earlgrey SoC used to evaluate MARVEL, their design size and number of bugs. Bugs were identified by comparing the buggy SoC with the open source implementation.

| Design IP | Description | Total Files | Design Files | Design LoC | # Bugs |
|---|---|---|---|---|---|
| adc_ctrl | Control/filter logic for dual A-to-D Converter. | 59 | 7 | 4159 | 2 |
| aes | Cryptographic accelerator for AES Standard. | 203 | 37 | 10425 | 9 |
| csrng | Supports deterministic (DRNG) and true random number generation (TRNG) compliant with FIPS and CC. | 69 | 12 | 5722 | 2 |
| entropy_src | FIPS and CC compliant entropy source used by csrng. | 91 | 20 | 7750 | 0 |
| hmac | SHA-2 hash-based authentication code generator. | 80 | 4 | 3613 | 3 |
| keymgr | The key manager implements the hardware component of the identities and root keys strategy of OpenTitan. | 75 | 14 | 5257 | 1 |
| kmac | Keccak-based message authentication code. | 202 | 16 | 7571 | 0 |
| lc_ctrl | Controller to manage product device lifecycle and associated functionality/access control. | 101 | 11 | 4027 | 3 |
| otbn | Co-processor for asymmetric crypto operations like ECC. | 440 | 24 | 8279 | 7 |
| otp_ctrl | Controller for the One-Time Programmable (OTP) memory. | 136 | 15 | 8612 | 10 |
| prim | Basic blocks used to implement the design; They are often technology-dependent and can have multiple implementations. | 501 | 164 | 14988 | 2 |
| tlul | Main system bus to interface the main processor core with peripherals; implements the TileLink protocol. | 87 | 21 | 2628 | 0 |

## 3.2 FRAMEWORK IMPLEMENTATION

We implemented MARVEL in Python, modeling the agentic framework using LangGraph. The implementation is open source at repository. MARVEL is fully automated. The flow is independent of the LLMs used, and can be transitioned to different models as they get released.

For our implementation, we use the same model for every agent, as this simplifies and constrains our design space. We considered Gemini 2.5 Pro, GPT-4.1, GPT-5 as possible model options. Gemini 2.5 Pro is Google's flagship model. GPT-4.1 is OpenAI's best non-reasoning model and provides the largest context window (1M), while GPT-5 is OpenAI's most advanced reasoning model. We set the default temperature of 0.15 for all models. Picking a small, non-zero value provides flexibility in the responses while ensuring that they remain conservative enough for the detailed tasks. We do not study prompt optimization; instead, we focus on the efficacy of the framework for security verification. For model selection, we performed small-scale experimentation on 3 IPs (adc_ctrl, aes, otp_ctrl). We selected these IPs because they cover I/O, crypto, and memory functionality. Figure 4 illustrates the number of reported security issues, number of actions, and runtime for both models. We classify every reported security issue as a **Bug** (a correct, actionable vulnerability in the RTL), **Warning** (a partially correct or security-relevant condition identified by the agent, but not an exploitable vulnerability), or **Hallucination** (an incorrect finding). Bugs are strict true positives, Warnings are soft true positive signals, and Hallucinations are false positives. We used

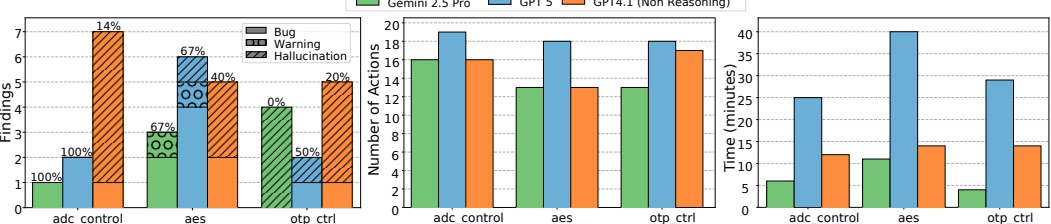

Figure 4: Results using reasoning (Gemini 2.5 Pro and GPT-5) and non-reasoning (GPT-4.1) models.

content analysis sessions, similar to prior work in software engineering (Catolino et al., 2019), to perform this classification. Two authors of this work independently reviewed each bug report and

the relevant design files. This includes RTL files, documentation, and test logs. When necessary, the OpenTitan repository (lowRISC contributors, 2023) was used as a golden reference. Then, a discussion was held to resolve any differences and reach a consensus. GPT-5 found 2× more Bugs in adc_ctrl and aes compared to Gemini 2.5 and GPT-4.1. Critically, GPT-5's precision is significantly higher than both other models across the board, and it hallucinates less. The action count is comparable for both models, with a slight increase across the board for GPT-5. This translates into higher runtime for GPT-5, due to the generation of reasoning tokens. Still, GPT-5 runtime remains reasonable, with a max of 40 minutes, which is a reasonable runtime given no human intervention is needed. Based on these observations, we use GPT-5 for our evaluation.

## 4 RESULTS

### 4.1 EVALUATION

**Overview**    For each of the 12 IPs, we evaluate the security properties and issues identified by MARVEL. The results are summarized in Table 2. All security properties identified by the supervisor agent were correctly formulated. We classify reported security properties and issues similarly to Section 3.2. The correctness of the identified security was determined by consulting the SoC documentation. The full list of identified issues, with descriptions taken from the generated reports and their respective classifications, is reported in Section A.2 and Section A.3. We additionally report precision, recall, and F1-score for each IP. These metrics provide a complementary view of performance: precision reflects how often reported findings are correct (Bug or Warning), recall captures how many true issues MARVEL uncovers, and F1 summarizes the tradeoff between the two. Performance varies across IPs, with MARVEL achieving perfect scores on three of them and worst scores on four. This suggests that hallucinations can lead the framework into unproductive analysis paths. All correctly identified bugs were also correctly localized. The run times span 18-53 minutes. The runtime depends on tool calls, with simulation and assertion verification being the two most time-consuming. The average cost per run is approximately $3. The kind of analysis carried out by MARVEL would require many hours for an experienced security engineer, highlighting the potential of LLMs to speed up hardware security evaluations. The findings are well distributed through all the IPs and the number of reported issues and hallucinations is relatively small, highlighting that MARVEL is effective at filtering the noisy outputs of the base tools. Assuming an engineering effort of 20 minutes per finding, the most expensive IP analysis would take 140 minutes, a small fraction of the typical man-months of security verification efforts. Moreover, integrating this analysis during design reduces the number of findings per run.

Table 2: Results summary for the 12 Design IPs in buggy OpenTitan earlgrey SoC. The Security Issues Localized includes only the correctly identified security issues (i.e., we ignore false positives).

| Design IP | Runtime [min.] | Security Properties Identified | Security Issues Identified | | | Prec. | Recall | F1 |
|---|---|---|---|---|---|---|---|---|
| | | | Bug | Warning | Hallucination | | | |
| adc_ctrl | 25 | 10 | 2 | 0 | 0 | 1.00 | 1.00 | 1.00 |
| aes | 40 | 10 | 4 | 1 | 0 | 1.00 | 0.44 | 0.62 |
| csrng | 37 | 9 | 0 | 2 | 4 | 0.00 | 0.00 | 0.00 |
| entropy_src | 27 | 16 | 0 | 3 | 1 | 0.00 | 0.00 | 0.00 |
| hmac | 26 | 10 | 3 | 0 | 0 | 1.00 | 1.00 | 1.00 |
| keymgr | 27 | 22 | 1 | 0 | 0 | 1.00 | 1.00 | 1.00 |
| kmac | 36 | 11 | 0 | 2 | 5 | 0.00 | 0.00 | 0.00 |
| lc_ctrl | 18 | 12 | 3 | 1 | 3 | 0.50 | 1.00 | 0.67 |
| otbn | 22 | 16 | 4 | 0 | 0 | 1.00 | 0.57 | 0.73 |
| otp_ctrl | 29 | 13 | 1 | 0 | 1 | 0.50 | 0.10 | 0.17 |
| prim | 22 | 10 | 1 | 1 | 1 | 0.50 | 0.50 | 0.50 |
| tlul | 53 | 9 | 0 | 4 | 3 | 0.00 | 0.00 | 0.00 |
| **Overall** | 362 | 148 | 19 | 14 | 18 | 0.51 | 0.49 | 0.50 |

**Supervisor Actions**    The initial prompt contains the path to the SoC base directory. From the example sequences of action in Section A.5, we see that the supervisor starts by exploring the SoC file structure and reading the documentation files to identify the security properties. Then it starts calling the available tools to check the identified security properties. The agent might inspect design files based on the tools' feedback to confirm and localize the bugs. Figure 5 shows the normalized and absolute number of actions performed by the supervisor agent. In every run,

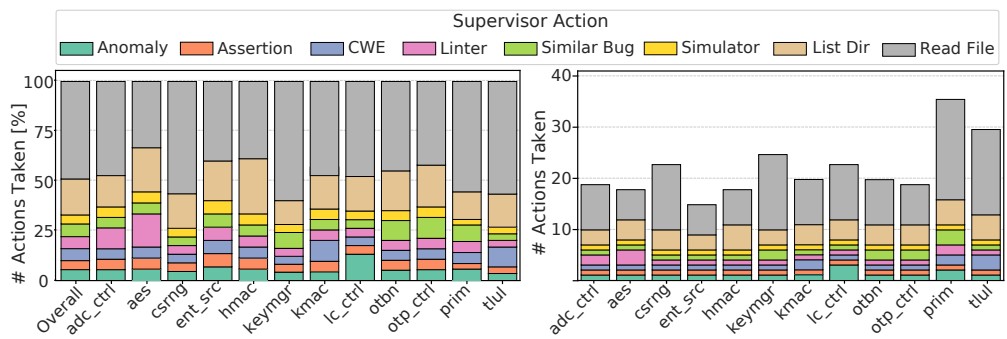

Figure 5: Normalized and Absolute Supervisor Action Distribution, Overall and for single IPs.

the supervisor agent calls each tool at least once. Prim has one of the lowest runtimes and the highest number of actions. This is due to the high number of file reads and directory listings; Prim has multiple basic blocks and has the most files in the SoC.

**Executor Agent Contribution** The roles of each agent in bugs reported by `MARVEL` are illustrated in Figure 6. Here, we focus on the actions that contributed to a result in the final report. Agents might not find any issues, in this case, the action does not contribute to the report. If the agent is used to determine and localize a confirmed security issue, it is described as the Determinator and Localizer. If it is used to identify the bug but is not the final determinator, it is a Helper. If it raises a warning, it is defined as a warner. If it is used in the flow of incorrectly identifying a security issue, it is defined as a False Identifier. More than one agent might count as Helper, Warner, or False Identifier. The CWE and Anomaly agents have the highest number of false identifications, with 8 and 11, respectively. These agents are not based on EDA tools.

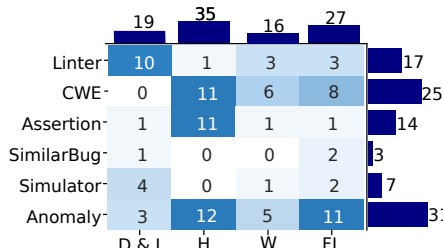

Figure 6: Roles of agents in bugs reported by `MARVEL`. **D**eterminator, **L**ocalizer, **H**elper, **W**arner, **F**alse **I**d.

**Executor Orchestration** We investigated the supervisor's ability to call executor agents based on file type and security objectives. Figure 7 shows the frequency of each file type–security objective pair as examined by the supervisor. The highest frequency is for the tuple access control and interface files (which implement most of the access control logic), followed by FSM security for FSM and control logic files. Tuples that do not make design sense, like entropy on interface modules, are never explored. The supervisor can accurately identify the security objectives, at least for some file categories. Section A.4 explains how we assigned security objectives and files to the respective classes.

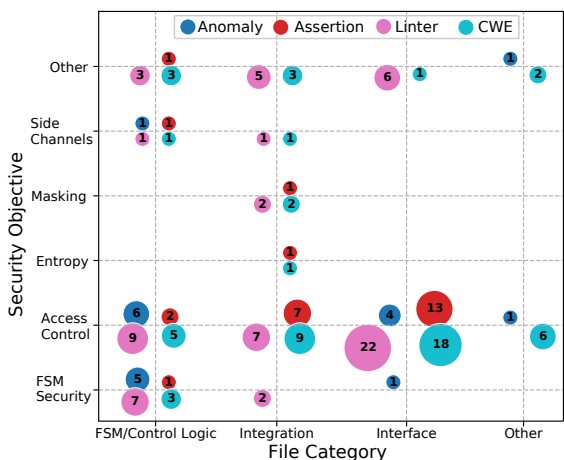

Figure 7: Agent activity frequency for each security objective and file category tuple for agents requiring a security objective and file category.

### 4.2 ARCHITECTURE ANALYSIS

**Benefits of Multi-Agent Architecture** We studied the benefits of the multi-agent, supervisor-executor framework by comparing it to a single-agent setup. For the single-agent setup, we used GPT-5 and exposed all tools through the tool-calling API. Results are illustrated in Figure 8. The agent's system prompt is shown in Section A.6. `MARVEL` is as good or better than the single agent at identifying security issues. On benchmarks where neither found any confirmed security issues, `MARVEL` raises warnings, whereas the single-agent setup provides only conclusive error reports.

**Executor Agent Ablation** This study explores the effectiveness of each executor agent. First, we ran the supervisor agent, excluding one executor agent at a time. Then we ran the supervisor agent with only one executor at a time. We did this analysis on a sub-

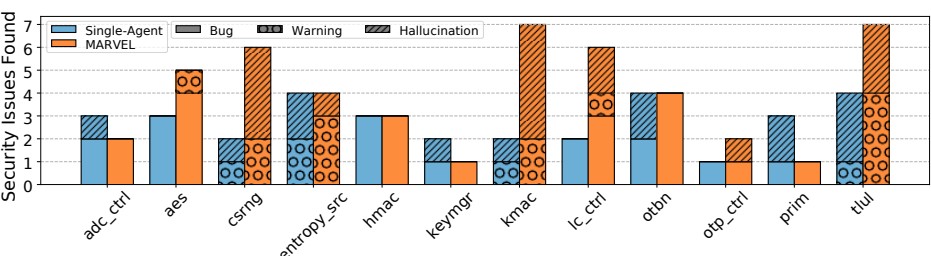

Figure 8: Comparison between MARVEL's multi-agent and single-agent setup on issues found.

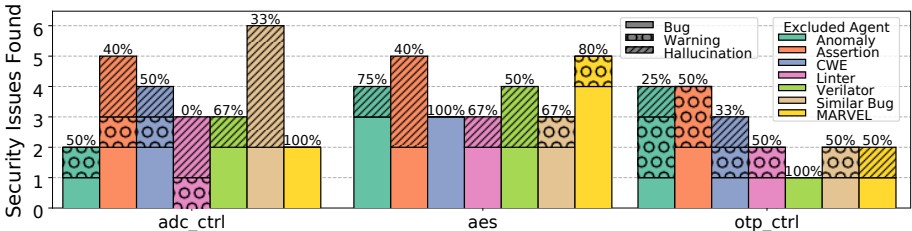

Figure 9: Ablation study: Supervisor with all Executors but one. % is for true positive ratios.

set of IPs. We selected the same subset used for model selection in Section 3.2. Figure 9 and Figure 10 show the number of bugs, warnings, and hallucinations reported, together with the True Positive to False Positive ratiofor the respective ablation studies. Both single executors and all-but-one runs have a lower True Positive ratio. Inspecting the actual bugs, we found that some bugs are less likely to be found when a specific executor is missing, like FSM bugs when the Linter agent is excluded. For the single executor we attribute the lower true positive ratio to the supervisor agent being

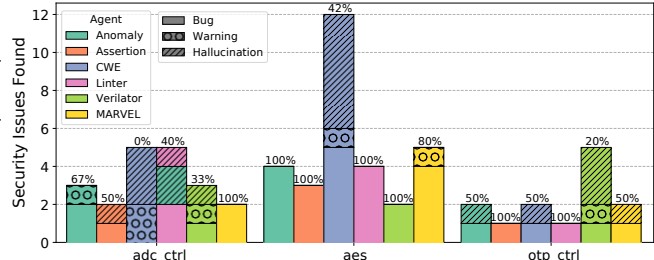

Figure 10: Ablation study: Supervisor plus a single Executor. % is for true positive ratios.

able to filter out false positives by using multiple tools. The different tools may identify the same simple problems, as the LLM can access the source and perform code analysis.

## 5 DISCUSSION

**Benefits** of the Supervisor-Executor architecture of MARVEL have been demonstrated through the interaction and coordination between agents. Based on the Design IP and their documentation, MARVEL was able to derive security objectives relevant to the IP accurately. The supervisor agent is then able to call on the executor agents according to the file type and security objective as shown in Figure 7. If one agent fails to provide helpful information or returns an error, the supervisor executes another agent until sufficient information is obtained to make a judgment about the violation of the security objective under consideration. Logs of the agentic flow revealed that MARVEL was able to iteratively improve calls to a tool until the syntactically correct call was made. An example can be found in Section A.5. The multiple calls in a row to the assertion agent on the same file are due to the tool call failing, and the supervisor agent attempting to correct the format and assertions. This highlights the benefit of an agentic approach, which allows the supervisor to correct its actions. A single-shot approach without the ability to iterate would result in unrecoverable failing tool calls. We demonstrated the use of hardware description language (HDL) specific tools, including VC SpyGlass Lint as the linter, VC Formal as the assertion tool, and Verilator as the simulator. LLMs can automate scripting these tools using templates, demonstrating that multi-agentic systems can be used for hardware code debugging.

**Limitations** From a security perspective, the strict true positive rate of 51% (discarding warnings) characterizes MARVEL's ability to identify actionable vulnerabilities. Considering warnings as weak positives raises the relaxed true positive rate to 64.7%, giving a more realistic picture of the system's utility for vulnerability triage. While a 51% precision is meaningful given the high cost of post-tape-out bugs, the 35.3% hallucination rate indicates substantial room for improvement. By open-sourcing our research, we aim to establish a foundation for systematically incorporating generative AI into security verification pipelines and driving these error rates down. Evaluating

the "quality" of the Supervisor's is difficult, as there is no optimal sequence. Our evaluation scope is limited to the efficacy and benefits of the multi-agent framework and tools. We used GPT-5 for each agent and did not explore multi-model composition or prompt optimization. Full multi-seed evaluations of MARVEL across all IPs are computationally costly, but its reliance on tool-grounded signals (lint, simulation logs, documentation) limits stochastic effects and makes single runs reliable, especially at temperature 0.15. We limit our benchmark to an OpenTitan-based SoC. The hardware security domain suffers from a lack of available data, and SoCs from the Hack@Event competitions are the best available source for real-world hardware bugs. The results observed should generalize to other designs, as we do not do any finetuning or design-specific optimizations. Finally, data leakage has minimal impact on MARVEL's effectiveness as the Hack@Date SoC and bug list are not public.

**Related Works** Prior work has explored LLMs for RTL debugging and vulnerability discovery, but important gaps remain. FLAG (Ahmad et al., 2025) used earlier LLMs for fault localization and showed that naive prompting can surface many candidates but suffers from very high false-positive rates. Surveys such as Saha et al. (2024) document promising LLM applications (insertion, verification, mitigation) but do not provide an end-to-end, tool-driven workflow for hardware. SV-LLM (Saha et al., 2025) moves toward an agentic setup and improves detection by fine-tuning on vulnerability examples, but it relies on heavy model specialization, does not expose its dataset or framework, and lacks tight runtime integration with verification tools. Self HW Debug (Akyash & Kamali, 2024) proposes an agentic flow to identify specific vulnerabilities. The framework is limited to five specific CWEs. Our work differs in substantial ways. First, MARVEL does not focus on explicit vulnerabilities. The security objectives are identified by the supervisor agent from the design documentation. The executor agents receive security objectives from the supervisor and adapt their execution to them using RAG. MARVEL is designed to be modular and retrieval-augmented (CWE and lint-tag retrievers), making it straightforward to extend with new executors or swap models without changing the orchestration logic. These features reduce false positives, improve actionable localization, and make our approach more practical for integration into verification pipelines than prior LLM-only or fine-tuned systems.

**Future Work** Research has started investigating prompt formation as an optimization problem (Pryzant et al., 2023). In this work, we focused on evaluating the effectiveness of our multi-agent framework and did not explore prompt optimization. MARVEL uses the same model for every agent to constrain the design space. Using different models may improve performance or cost. A natural extension of MARVEL would be to add more executor agents. This could include using other techniques used for RTL security bug detection, like Information Flow Tracking (Hu et al., 2021) and Fuzzing (Rostami et al., 2024). Another direction for exploring multi-agent systems for RTL security bug detection would be to employ an architecture where executor agents can communicate directly with each other. Alternatively, a multiple hierarchy could have agents between the supervisor and executor agents that are primed to use broad categories of tools. A Static Analysis agent could control flow between the Linter, Assertion, and Anomaly agents, and a Known Bug agent could include the CWE and Similar Bug agents. Other improvements could include a separate localizer agent that uses non-LLM-based techniques to localize bugs based on information from the Supervisor Agent. These could include embedding and keyword searches.

## 6 CONCLUSION

We introduced MARVEL, a multi-agent LLM framework for automated detection of RTL security vulnerabilities. Our results show that each agent contributes to the security analysis at least as a helper. MARVEL reported 51 potential security issues, of which we manually evaluated 19 as confirmed issues and 14 as relevant security warnings. All 19 confirmed issues were also correctly localized to the relevant file and lines. To quantify overall performance, we report strict precision, recall, and F1-score as summarized in Table 2. MARVEL achieves an overall precision of 0.51, a recall of 0.49, and an F1-score of 0.50. The per-IP metrics exhibit a highly skewed distribution: some IPs contain few real issues and achieve perfect scores (1.00 precision, recall, and F1). In contrast, others present more challenging conditions and yield 0–0–0 outcomes. This reflects the heterogeneous nature of the designs and highlights that MARVEL performs well when actionable issues are present, but remains sensitive to design complexity and noise in more demanding settings. Our work highlights LLMs' potential to speed up hardware security evaluations. Yet they augment, not replace, human expertise for key applications, as experienced professionals must conduct the final assessment and critical decisions. While promising, false positives remain a limitation; future work will focus on reducing hallucinations and expanding executor capabilities.

## 7 REPRODUCIBILITY STATEMENT

Our code and results are available to reviewers through an anonymized repository. We note that our framework, although open-source, utilizes proprietary tools. To run it successfully, users need licenses for these tools. Unfortunately, we cannot provide the source code of our benchmark, as the Hack@DATE 2025 SoC has not been made publicly available by the organizers of the competition. Individuals can reach out to the organizers to ask for the SoC source code.

## 8 ETHICAL CONSIDERATIONS

Ethical considerations must be taken into account when working with cybersecurity. The possibility of malicious use of the tool should be taken into consideration. This includes both the use of methods to find vulnerabilities with harmful intent and changes to system prompts that may allow the objective to be changed from bug detection to bug insertion. In both scenarios, the user would need to access design files, which in the hardware domain are accessible by trusted employees. In the hardware domain, these scenarios are less of a concern than in the software domain.

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

## A  TECHNICAL APPENDICES AND SUPPLEMENTARY MATERIAL

### A.1  ILLUSTRATIVE EXAMPLES FOR EXECUTOR AGENTS

This section includes examples of the executor agents' detection of security issues not covered in the main pages of this manuscript.

#### A.1.1  EXAMPLE FOR LINTER AGENT

---

**Instruction to Linter Assistant from Supervisor Agent**

```
Are there FSM related security concerns with the provided RTL:
<RTL source code> <Design filepath> <Top module>
Identify relevant lint tags for the RTL to be checked for.
Then run the tests using the linting tool.
From the output of the linting tool, determine if there are security issues in
the RTL.
```

**Tool call from Linter Assistant**

```
Tool Call:  retrieve_relevant_lint_tags
Arguments:  query:  FSM security
```

**Message from Lint Tags Retriever to Linter Assistant**

```
LINT_FSM_UNREACHABLE_STATES: Do not have unreachable states in FSMs
LINT_FSM_WITHOUT_EXIT_STATE: Do not have states without exit in FSMs
...
```

**Tool call from Linter Assistant**

```
Tool Call:  lint_checker_tool
Arguments:
- design_filepath:  <path-to-adc-ip>/adc_ctrl_fsm.sv
- top_module:  adc_ctrl_fsm
- lint_tags:  [...  'MissingFsmStateTransition',
'LINT_FSM_WITHOUT_INITIAL_STATE', 'LINT_FSM_UNREACHABLE_STATES',...]
```

**Message from Lint Checker Tool to Linter Assistant**

```
Successfully executed lint checker tool
Report Summary:
Tag :  LINT_FSM_UNREACHABLE_STATES
Description :  State 'LP_PWRUP' is unreachable
LineNumber :  306
Statement :  LP_PWRUP: begin
...
```

**Message from Linter Assistant to Supervisor Agent**

```
The linting tool has identified several security concerns related to the
finite state machine (FSM) in the provided RTL code.  Here are the details
of the findings:
**LINT_FSM_UNREACHABLE_STATES**:
- **Description**:  State 'LP_PWRUP' is unreachable.
- **Severity**:  Error
- **Line Number**:  306
- **Details**:  This state cannot be reached from any other state, which can
lead to unexpected behavior in the FSM.
```

---

Figure 11: Example linter agent flow for Improper FSM on ADC Control FSM module. Linter assistant receives instructions to find FSM security concerns. It calls the lint tags retriever to obtain FSM-related tags and then calls the lint checker to execute the lint tags on the ADC Control FSM. From the output of the checker tool (LINT_FSM_UNREACHABLE_STATES tag violated), the assistant determines that state LP_PWRUP is unreachable and points to line 306 as the bug location.

### A.1.2 EXAMPLE FOR ASSERTION AGENT

---

**Instruction to Assertion Assistant from Supervisor Agent**

Are there security concerns related to the wipe secret operation in the provided RTL?
Is the wipe secret operation occurring at the appropriate conditions?
`<RTL source code> <Design filepath> <Top module>`
Form relevant system verilog assertions for the RTL to be checked for.
These system verilog assertions should be in a dict with the key as the name and the value as the assertion string.
An example assertions object to be sent to assertion checker tool is:
`<Example Assertions>`
From the output of the assertion checker tool, determine if there are security issues in the RTL.
If there are no falsified assertions in the output from the assertion checker tool, that means there are no verified security issues in the RTL.

---

**Tool call from Assertion Assistant**

```
Tool Call:  assertion_checker_tool
Arguments:
- design_filepath:  <path-to-hmac-ip>/hmac_reg_top.sv
- top_module:  hmac_reg_top
- clock_signal:  clk_i
- reset_signal:  rst_ni
- reset_active:  0
- assertions:  {'assertion_wipe_secret':  "property p_wipe_secret;
@(posedge clk_i) disable iff (!rst_ni) (wipe_secret_we && !reg_error)
|-> (reg2hw.wipe_secret.q == 32'h0); endproperty assertion_wipe_secret:
assert property (p_wipe_secret);", 'assertion_wipe_secret_condition':
'property p_wipe_secret_condition; @(posedge clk_i) disable iff (!rst_ni)
(wipe_secret_we && !reg_error) |-> (addr_hit[8] && reg_we); endproperty
assertion_wipe_secret_condition:  assert property (p_wipe_secret_condition);'}
```

---

**Message from Assertion Checker Tool to Assertion Assistant**

Successfully executed:  assertion checker tool
Output:  There are no falsified properties.

---

**Message from Assertion Assistant to Supervisor Agent**

The assertion checker tool did not find any falsified assertions, which means there are no verified security issues related to the "wipe secret" operation in the provided RTL. The "wipe secret" operation is occurring under the appropriate conditions, and the secret is being wiped correctly when the operation is triggered without any register errors.

---

Figure 12: Assertion Agent flow for wipe secret security objective for HMAC Register module.

### A.1.3 EXAMPLE FOR CWE AGENT

**Instruction to CWE Assistant from Supervisor Agent**

```
Are there security concerns related to hardware child block instantiations in
the provided RTL?
<RTL source code>
Identify the CWE relevant to the security issue for the given RTL.
Obtain details of the CWE.
Then determine if there are security issues relevant to the identified CWE in
the RTL.
```

**Tool call from CWE Assistant**

```
Tool Call:  cwe_details_retriever_tool
Arguments:  security_issue:  hardware child block instantiations
```

**Message from CWE Details Retriever to CWE Assistant**

```
CWE-1276:  Hardware Child Block Incorrectly Connected to Parent System
Signals between a hardware IP and the parent system design are incorrectly
connected causing security risks.
<Extended Description> <Example 1> <Example 2>
```

**Message from CWE Assistant to Supervisor Agent**

```
In the provided RTL code, we need to ensure that all child block
instantiations are correctly connected to the parent system.  Let's examine
the instantiations and connections:
1.  **prim_blanker Instantiations**:
- 'prim_blanker #(.Width(WLEN)) u_operand_a_blanker' and 'u_operand_b_blanker'
are instantiated with the enable signal hardcoded to '1'b1'.  This is correct
if the intention is to always enable these blankers, but it should be verified
against the design specifications.
```

Figure 13: Example CWE Agent flow on OTBN MAC Bignum module

### A.1.4 EXAMPLE FOR SIMILAR BUG AGENT

**Instruction to Similar Bug Assistant from Supervisor Agent**

```
This bug was previously found in another file of this design:  DIP_CLEAR:
data_in_prev_d = data_in;
Please check the RTL code in the file <path-to-IP>/ip/aes/rtl/aes_cipher_core.sv
for similar bugs.
```

**Tool call from Similar Bug Assistant**

```
Tool Calls:  similar_bug_tool
Args:
- bug:  DIP_CLEAR: data_in_prev_d = data_in;
- ip_file:  <path-to-IP>/ip/aes/rtl/aes_cipher_core.sv
```

**Message from Similar Bug Tool to Similar Bug Assistant**

```
Successfully executed:  similar_bug_tool// Found similar bug in
<path-to-IP>/aes/rtl/aes_cipher_core.sv at line 585:  .data_out_clear_i (
data_out_clear_i ),
KEY_DEC_CLEAR: key_dec_d = key_expand_out;
Found similar bug in <path-to-IP>/aes/rtl/aes_cipher_core.sv at line 442:
KEY_FULL_CLEAR: key_full_d = key_expand_out;
Found similar bug in <path-to-IP>/aes/rtl/aes_cipher_core.sv at line 251:
STATE_CLEAR: state_d = add_round_key_out;
Found similar bug in <path-to-IP>/aes/rtl/aes_cipher_core.sv at line 234:
logic[3:0][3:0][7:0] data_in_mask;
Found similar bug in <path-to-IP>/aes/rtl/aes_cipher_core.sv at line 249:
STATE_INIT: state_d = state_init_i;
<continues>
```

**Tool call from Similar Bug Assistant**

```
To better understand the context, I'll read the file to examine these lines
more closely:
Tool Calls:  read_file_with_line_numbers
Args:
- file_path:  <path-to-IP>/ip/aes/rtl/aes_cipher_core.sv
```

**Message from Read File Tool to Similar Bug Assistant**

```
Successfully executed:  read_file_with_line_numbers <file-content>
```

**Message from Similar Bug Assistant to Supervisor Agent**

```
After carefully analyzing the results, I found a few lines that might be
similar to the original bug, but they are not exact matches:
1.  Line 461:  `KEY_DEC_CLEAR: key_dec_d = key_expand_out;`
- This is in a key decryption context, which is different from the original
bug.
2.  Line 442:  `KEY_FULL_CLEAR: key_full_d = key_expand_out;`
- This is in a full key context, which is different from the original bug.
3.  Line 251:  `STATE_CLEAR: state_d = add_round_key_out;`
- This is in a state context, which is different from the original bug.
None of these lines are exact matches to the original bug `DIP_CLEAR:
data_in_prev_d = data_in;`.  The lines found are similar in structure (using
a clear/reset state and assigning a value), but they are in different contexts
(key, state) and do not involve `data_in_prev_d`.

Therefore, I cannot confidently identify any buggy lines that are direct
matches to the original bug in this file.
```

Figure 14: Example Similar Bug Agent flow for a similar bug previously found in aes_core.sv.

### A.1.5   EXAMPLE FOR ANOMALY AGENT

**Instruction to Anomaly Assistant from Supervisor Agent**

```
Are there security concerns related to the wipe secret operation in the
provided RTL?
<RTL source code> <Design filepath> <Top module>
Use the anomaly detector tool to identify lines in the design RTL that are
anomalous.
Then determine whether the identified anomalous line(s) represent a security
issue or not.
```

**Tool call from Anomaly Assistant**

```
Tool Call:  anomaly_detector_tool
Arguments:
- design_filepath:  <path-to-hmac-ip>/hmac_reg_top.sv
```

**Message from Anomaly Detector Tool to Anomaly Assistant**

```
Successfully executed:
Anomaly Detector tool
Output:
Here are clusters of similar verilog constructs in the RTL file:
Cluster 0:
assign status_re = addr_hit[6] & reg_re & !reg_error; assign wipe_secret_we
= addr_hit[8] & reg_we & reg_error; assign wipe_secret_wd = reg_wdata[31:0];
assign key_0_we = addr_hit[9] & reg_we & !reg_error;
Cluster 1:
lines in cluster 1
Cluster 2:
lines in cluster 2
```

**Message from Anomaly Assistant to Supervisor Agent**

```
The anomaly detection tool has identified the following line as potentially
anomalous:
`
assign wipe_secret_we = addr_hit[8] & reg_we & reg_error;
`
This line is responsible for enabling the write operation to the `wipe_secret`
register.  The anomaly here is that the write enable (`wipe_secret_we`) is
conditioned on `reg_error` being true, which is unusual because typically,
write operations should not proceed when there is an error condition.
Security Implications
Recommendation
```

Figure 15: Anomaly Agent flow for wipe secret security objective for HMAC Register Top module.

A.1.6    EXAMPLE FOR SIMULATOR AGENT

---

**Instruction to Simulator Assistant from Supervisor Agent**

```
Run a security analysis on the aes IP.
Inspect the logs of failing runs and determine if there are security issues in
the RTL. If any security issues are found, provide a detailed explanation of
the issue and its location in the RTL code.
```

**Tool call from Simulator Assistant**

```
Tool Call:  run_verilator_tests
Arguments:
- ip:  aes
```

**Message from Verilator Tool to Simulator Assistant**

```
Successfully executed:
Verilator tool
Output:
...
//sw/device/tests/crypto:aes_masking_off_test_sim_qemu_rom_with_fake_keys
FAILED in 4.3s
<path-to-log>/aes_masking_off_test_sim_qemu_rom_with_fake_keys
//sw/device/tests/crypto:aes_kwp_sideload_functest_sim_qemu_rom_with_fake_keys
FAILED in 3.1s
<path-to-log>/aes_kwp_sideload_functest_sim_qemu_rom_with_fake_keys/test.log
//sw/device/tests/crypto:aes_sideload_functest_sim_qemu_rom_with_fake_keys
FAILED in 3.1s
<path-to-log>/aes_sideload_functest_sim_qemu_rom_with_fake_keys/test.log

Executed 11 out of 45 tests:  8 tests pass, 1 fails to build, 11 fail locally
and 25 were skipped.
There were tests whose specified size is too big.  Use the
-test_verbose_timeout_warnings command line option to see which ones these are.
```

**Tool Call from Simulator Assistant**

```
I'll examine one of the failing test logs to identify potential security
issues.  Let's look at the masking off test, which seems particularly relevant
for security analysis:
Tool Call:  read_file
Arguments:
- file_path:  <path-to-log>/aes_masking_off_test_sim_qemu_rom_with_fake_keys
```

**Message from Read File Tool to Similar Bug Assistant**

```
Successfully executed:  read_file <file-content>
```

**Message from Similar Bug Assistant to Supervisor Agent**

```
Based on the analysis of the failing tests, I've identified several security
issues in the AES IP:
Security Issues Found:
1.  **Masking Protection Failure (Side-Channel Vulnerability)**
- The `aes_masking_off_test` shows a failure in the CSRNG (Cryptographically
Secure Random Number Generator) interface, specifically with the reseed
counter.
- This indicates that the masking countermeasures designed to protect against
side-channel attacks are not functioning correctly.
- Side-channel attacks could potentially extract secret keys by monitoring
power consumption or electromagnetic emissions during encryption operations.
2.  **PRNG Reseed Mechanism Failure (Cryptographic Weakness)**
- The `aes_prng_reseed_test` reveals that the AES module continues to operate
even when entropy is disabled, which is a serious security issue.
- The test expected the AES operation to halt when entropy was disabled (at
block 32), but it continued processing until block 63.
- This means the AES implementation might be using predictable or weak
randomness for its operations, making it vulnerable to cryptanalysis attacks.
3.  **Stall Mechanism Failure (Potential Fault Injection Vulnerability)**
- The `aes_stall_test` shows that the AES module does not stall as expected
when back pressure is applied.
- This could potentially be exploited in fault injection attacks, where an
attacker manipulates the timing or operation of the cryptographic module to
extract sensitive information.
- The failure indicates that the AES module might not properly handle resource
contention, which could lead to unpredictable behavior or information leakage.
```

Figure 16: Example Simulator Agent for AES.

## A.2 REPORTED ISSUES

Table 3: Reported Issues.

| Reported Issue ID | IP | Description (As produced by LLM) |
|---|---|---|
| 1 | aes | The read-data mux returns actual key_share0 values to software and, in the default case, returns key_share0[0] even for invalid addresses. This violates the documented policy that key registers are write-only and creates a critical confidentiality leak path, including under error responses. |
| 2 | aes | The read mux maps KEY_SHARE1_0..3 addresses to DATA_IN readback and maps DATA_IN addresses to zero. This is a functional and policy violation (KEY_SHARE1 is supposed to be write-only; DATA_IN is also write-only per spec). It can also expose plaintext input at unexpected addresses. |
| 3 | aes | DIP_CLEAR case assigns data_in_prev_d = data_in instead of pseudorandom clearing data (prd_clearing_data). This defeats the intended secure wipe (DATA_REG.SEC_WIPE) of the previous-input register, leaving deterministic/correlated values where PRD should be used. |
| 4 | aes | Reset branch is conditioned on data_out_we != SP2V_HIGH: if reset asserts while data_out_we equals SP2V_HIGH, the reset path does not clear data_out_q. This risks retaining prior ciphertext in software-readable registers across reset. |
| 5 | aes | Some unique case/if-else constructs lack default 'x assignments or ternary usage that promote X-propagation in simulation. This can mask illegal encodings/fault scenarios during verification, reducing confidence in control-path hardening coverage. |
| 6 | adc_ctrl | Low-power sleep state (LP_SLP) has no exit transition. When the wakeup timer reaches its programmed threshold, the FSM only clears the counter and does not transition to LP_PWRUP (or back to sampling). This causes a permanent low-power sleep loop. LP_PWRUP is therefore unreachable. |
| 7 | adc_ctrl | Threshold computation underflows when software programs zero into adc_lp_sample_ctl.lp_sample_cnt or adc_sample_ctl.np_sample_cnt. The RTL computes lp_sample_cnt_thresh = cfg_lp_sample_cnt_i - 1 and np_sample_cnt_thresh = cfg_np_sample_cnt_i - 1 without clamping or HW enforcement. If SW writes 0 (despite the spec "must be 1 or larger"), the threshold wraps to 0xFF/0xFFFF, delaying debounced matches drastically and potentially preventing timely detection. |
| 8 | otp_ctrl | Hidden counter-based trigger ("Predict Mechanism") bypasses DAI access-control locks. A 2-bit saturating counter lock_cnt is incremented on otp_access_grant and, once it equals Predictor_Mask (2'b11), it is OR'ed into every critical access check for DAI read/write/scramble/digest paths. This permits reads/writes (including to secret partitions and digest regions) even when read_lock/write_lock are asserted, undermining multi-bit encoded (MUBI) access controls and documented partition policies. Due to operator precedence, some write-path conditions can allow issuing requests without part_sel_valid when the bypass is active. This is a classic stealthy hidden trigger/backdoor pattern. |

| 9 | otp_ctrl | TL-UL SW window OOB read acknowledged without rerror. When TL-UL address doesn't match any partition (tlul_part_sel_oh == 0), tlul_oob_err_q is set; tlul_gnt and tlul_rvalid are asserted, but tlul_rerror is left at '0 (success) and tlul_rdata at '0. This silently treats OOB reads as successful zero-data reads instead of erroring out, contrary to documentation that out-of-bounds reads should error. It can mask misuse and weaken software-side robustness checks. |
|---|---|---|
| 10 | csrng | GENBITS valid and FIPS status are always exposed to SW via hw2reg.genbits_vld.{genbits_vld,genbits_fips} regardless of OTP gate and SW_APP_ENABLE. Only the data path hw2reg.genbits.d is gated by (sw_app_enable && efuse_sw_app_enable[0]). Software can observe activity (valid) and FIPS status even when data reads are disabled by OTP or policy. |
| 11 | csrng | - Description: cmd_result_ack_rdy = (cmd_blk_select && state_db_wr_req_rdy) && ctr_drbg_gen_req_rdy; This couples the ack path for non-GEN commands (Instantiate/Reseed/Update/Uninstantiate) to the generate-path ready signal. Backpressure or blockage on the GEN path can delay acks for non-GEN operations, increasing DoS surface. |
| 12 | csrng | Writing ERR_CODE_TEST selects an error index that feeds many error sum signals and event_cs_fatal_err. This is intended for testing but, absent lifecycle gating/locking in production, permits SW-triggered fatal alerts/interrupts (DoS). REGWEN can lock writes if firmware clears it; however, there is no lifecycle-based hardware enforcement here. |
| 13 | csrng | The CS bus consistency check compares only the lower 64 bits of 128-bit genbits to detect repeats. An attacker manipulating only upper 64 bits could evade detection; benign repeats on upper half won't be flagged. |
| 14 | csrng | acmd_flag0_pfa = mubi4_test_invalid(flag0_q). The invalid check applies to the registered flag0 and only when INS/RES capture it. Invalid encodings on other commands are ignored. Likely intended, but you may also wish to check the incoming field at SOP for INS/RES. |
| 15 | csrng | Detailed internal FSM state and per-instance reseed counters are exposed to SW as RO debug/status and are not gated on lifecycle. If unprivileged SW can read these CSRs, they may leak internal activity patterns. |
| 16 | entropy_src | When firmware sets ES_ROUTE to true (route to SW), esfinal FIFO pops to SW only when swread_done is asserted. If OTP disables SW CSR reads (otp_en_entropy_src_fw_read = False), es_data_reg_rd_en remains false, the SW read index is continuously cleared, and swread_done never asserts. Meanwhile, ACK SM is masked off from popping to HW when ES_ROUTE is true. Result: esfinal FIFO eventually fills and seeds are dropped silently, starving the HW entropy consumer (availability degradation/DoS). This is a privilege/policy inconsistency between the control (routing) and the data access permission (OTP).

- Security aspect affected: Access control consistency and availability (DoS). |

| | | |
|---|---|---|
| 17 | entropy_src | If esfinal FIFO is full and a new seed arrives, the seed is dropped. This is not surfaced to firmware via a dedicated status or recoverable alert (documentation explicitly says firmware is not informed). While this may be acceptable by design, it reduces diagnosability and can be abused in conjunction with routing/policy to degrade availability. |
| 18 | entropy_src | Unlike other critical MUBI controls, rng_fips uses mubi4_test_true_loose for its functional enable while still raising a recoverable alert on invalid encoding. Under certain single-bit faults, a "loose-true" could spuriously assert rng_fips_o in the same cycle an invalid alert is raised. This is less fail-closed than using strict decoding. |
| 19 | entropy_src | Most MUBI controls are synchronized and fanned out (prim_mubi4_sync) to separate "functional enable" vs "invalid" evaluation paths, improving spatial/fault hardening. threshold_scope directly uses the raw MUBI value for both pfe/pfa without synchronization/fanout, which is inconsistent with the rest of the design's hardening pattern. |
| 20 | hmac | WIPE_SECRET write-enable is asserted only when there is a register error: 

 assign wipe_secret_we = addr_hit[8] & reg_we & reg_error; 

 All other CSRs use "& !reg_error". As a result, a normal write (no error) never triggers the wipe. In hmac.sv the wipe controls are driven by reg2hw.wipe_secret.qe; with this bug, secure wipe never occurs under normal conditions. |
| 21 | hmac | - addr_hit[8] (WIPE_SECRET read) returns reg2hw.key[0].q (the first key word). 

 - addr_hit[9] (KEY_0 read) returns reg2hw.key[1].q. 

 These CSRs are supposed to be write-only (and WIPE_SECRET read should not expose secrets). Returning key data violates key confidentiality. |
| 22 | hmac | The default branch of the read mux returns key material: 

 default: reg_rdata_next = reg2hw.key[2].q; 

 On any address miss (or unexpected mux path), driving key data on the bus is a severe leak. Even if the bus flags an error, rdata is often still observable to software/debug infrastructure. |
| 23 | keymgr | - The data enable FSM's default branch handles illegal/unexpected state encodings by forcing state_d = StCtrlDataDis (fail-closed) but does not assert fsm_err_o. The intended error assertion is present but commented out: 

 //fsm_err_o = 1'b1; 

 state_d = StCtrlDataDis; 

 - fsm_err_o is initialized to 0 at the beginning of the always_comb and is never set to 1 anywhere in the module. Consequently, any illegal state or encoding corruption will not raise the FSM error signal. Other keymgr FSMs do assert their error outputs in the default branch, making this module an outlier that weakens the fault detection posture. |

| 24 | kmac | rand_valid_o is asserted in StRandReset with dummy/predictable data prior to true seeding. rand_data_q resets to a fixed RndCnstBufferLfsrSeed, and in StRandReset the FSM sets rand_valid_set=1 until SW asserts entropy_ready and the module transitions to a proper entropy mode. Consumers may see "valid" randomness before a true seed is established if system sequencing is incorrect. |
| 25 | kmac | The entropy module uses a formal ASSUME that consumers never drive rand_update_i or rand_consumed_i unless rand_valid_o is asserted (except immediately after seed_done). There is no hardware enforcement at the interface to block misuse; correctness relies on integration and verification. Security aspect: Control integrity of randomness consumption; potential misuse if upstream violates the assumption (could use stale or dummy randomness). |
| 26 | kmac | ENTROPY_SEED is not gated by CFG_REGWEN (cfg_regwen_qs). Unlike most sensitive CSRs, entropy_seed_we is allowed whenever addressed (subject to integrity and address checks). In sw_mode this is required to initialize the PRNG, but if write access is not privilege-restricted at the SoC level, untrusted software could attempt to influence the PRNG reseed process during configuration windows. |
| 27 | kmac | Verilator test kmac_error_conditions_test_sim_verilator indicates missing recoverable alert on shadowed register update mismatch (test expected status.alert_recov_ctrl_update_err to set). RTL appears to wire shadowed_update_err through to alerts[0] and to a sticky status bit that clears on err_processed. The discrepancy suggests a potential propagation/timing/gating issue (e.g., REGWEN gating masking the shadowed write sequence) or an environment/test harness issue; requires further waveform-based investigation. |
| 28 | kmac | When a hardware application interface is active (mux_sel != SelSw), SW writes to MSG_FIFO are accepted by the TL adapter (sw_ready_o defaults to 1) but silently dropped at the KMAC data mux. The module reports ErrSwPushedMsgFifo via ERR_CODE, but the TL transaction will appear successful to SW. While intended, this can allow a misbehaving or malicious SW client to generate back-to-back dropped writes, potentially contributing to system-level DoS or confusion if software does not check ERR_CODE and status/interrupts. |
| 29 | kmac | Exposed SW controls can reduce hardening: entropy_fast_process (reuses entropy except in key block; doc warns SCA leakage), en_unsupported_modestrength (enables unsupported mode/strength combos), and msg_mask (disables message masking). If untrusted software can set these, they can degrade protections. |
| 30 | kmac | EDN wait timer disable semantics: non-zero wait_timer_limit is latched on timer_update; changing to zero mid-request does not take effect until timer_update. If a nonzero limit expires, module enters error handling (ErrWaitTimerExpired). This is documented as intended but can lead to spurious DoS if software tries to "poke" the timer mid-transaction to avoid an error. |

| 31 | lc_ctrl | Token verification compares only the lower 32 bits of the 128-bit hashed token in all three checks (TokenHashSt, TokenCheck0St, TokenCheck1St). This reduces authentication strength from 128 bits to 32 bits in total, significantly weakening the brute-force resistance of the lifecycle token mechanism. |
| --- | --- | --- |
| 32 | lc_ctrl | LcStProd is erroneously included in the "test unlocked" decode block, enabling DFT_EN, NVM_DEBUG_EN, HW_DEBUG_EN, and setting keymgr diversification to "TestUnlocked" for production state. A separate "Enable production functions" block exists with the correct production policy, resulting in conflicting behavior. |
| 33 | lc_ctrl | Functional code in IdleSt allows asserting lc_clk_byp_req in several states including LcStDev and LcStProd if use_ext_clock_i is set, but the assertion explicitly forbids clock bypass in DEV/PROD/PROD_END. This is a design inconsistency. |
| 34 | lc_ctrl | Volatile RAW unlock path bypasses KMAC and directly compares unhashed_token_i to RndCnstRawUnlockTokenHashed (naming suggests digest, though comparison domain may be intended). Even if intended for test chips only (gated by SecVolatileRawUnlockEn), it's a sensitive unlock path that must be disabled in production. |
| 35 | lc_ctrl | The DEV-state comment says "access to the isolated flash partition is disabled." However, lc_iso_part_sw_wr_en is set to On in LcStDev (read remains Off). This contradicts the comment and may violate intended policy depending on spec. |
| 36 | lc_ctrl | TAP path lacks full TL-UL bus integrity; only a WE one-hot checker feeds into fatal_bus_integ_error. Security relies on life-cycle gating elsewhere (HW_DEBUG_EN/DFT_EN isolation via pinmux). This is acceptable by design but must be enforced system-wide in PROD states. |
| 37 | lc_ctrl | alert_test CSRs allow SW/TAP to trigger fatal alerts. This can be used as a local DoS by any agent with write access. Typically acceptable for testing, but consider life-cycle gating in production to reduce DoS surface. |
| 38 | otbn | Secure wipe request is hard-tied low. The intended handshake to request post-execution secure wipe is commented out and replaced by a constant 0.
- Code:
//assign secure_wipe_req_o = start_secure_wipe \| secure_wipe_running_q;
assign secure_wipe_req_o = 1'b0;
- Impact: Disables the post-execution secure wipe mechanism described in the spec. May allow transitions to Locked or Idle without performing mandatory wipes, potentially leaving sensitive data in WDRs/GPRs/ISPRs/DMEM/IMEM. Violates documented "Reaction to Fatal Errors" and "Secure Wipe" behavior. |

| 39 | otbn | LSU address SCA blanking bypassed and multiple drivers on lsu_addr_o. The address is blanked via prim_blanker (SEC_CM: DATA_REG_SW.SCA), then immediately overridden by a second assignment to the raw address.

- Code:

prim_blanker ...  u_lsu_addr_blanker (.in_i (lsu_addr), .en_i (lsu_addr_en_predec_i), .out_o(lsu_addr_blanked));

assign lsu_addr_o = lsu_addr_blanked;

assign lsu_addr_o = lsu_addr; // overrides blanked value

- Impact: Disables the LSU address blanking countermeasure, increasing side-channel leakage of memory access patterns and weakening the intended redundancy with predecode enable. Also introduces multiple continuous drivers (illegal/unsafe). |
| 40 | otbn | ISPR write commit forced always-on and multiple drivers. The proper commit gating (ispr_wr_insn & insn_executing) is immediately overridden with a constant 1.

- Code:

assign ispr_wr_commit_o = ispr_wr_insn & insn_executing;

assign ispr_wr_commit_o = 1'b1;

- Impact:  Commits ISPR writes unconditionally, even on stalled/error cycles. This undermines execution/error gating and can corrupt internal state or leak/destabilize control/flags, undermining integrity and fault hardening. |
| 41 | otbn | Bus read-data blanking enable forced high for both IMEM and DMEM TL-UL windows.  The code comments state blanking should occur during core operation, dummy responses, and locked state; however, en_d is tied to 1'b1, disabling blanking.

- Code:

// SEC_CM: DATA_REG_SW.SCA (comments say to blank during core operation/lock/dummy)

assign imem_rdata_bus_en_d = 1'b1;

...

assign dmem_rdata_bus_en_d = 1'b1;

- Impact:

- When OTBN is busy or locked, bus reads are supposed to return zero.  With en=1, the blanker will pass internal memory data to the bus path, violating "reads return zero when busy/locked" and weakening side-channel protections and data confidentiality on the system bus.

- Likely violates multiple in-RTL assertions (e.g., reads-as-zero when locked). |

| 42 | prim | Undriven error_s suppresses mismatch detection; shadowed register double-write integrity is disabled. |
| 43 | prim | Reset-domain crossing can cause spurious dst_req; with in-flight txn_bits_q, this can assert unintended destination write/read/regwen strobes on reset release. |
| 44 | prim | For Status-type interrupts, INTR_STATE must be RO/external; passthrough path uses CSR.q for intr_o instead of live status, potentially allowing SW to momentarily affect intr_o if misconfigured; adds one-cycle latency in passthrough. |
| 45 | tlul | The adapter intentionally omits a base address/size check. SRAM address is derived by slicing AHB/TL address bits: addr_o = tl_i_int.a_address[DataBitWidth+:SramAw]. If the crossbar routes a larger window than the actual SRAM, higher address bits are discarded, causing aliasing. Requests beyond the SRAM size may wrap and access unintended rows. |
| 46 | tlul | - Both EnableDataIntgGen and EnableDataIntgPt can be disabled, leaving no data integrity on the path.

- DataXorAddr protection (XORing address with data in the memory controller) is only handled in the passthrough integrity path. If DataXorAddr=1 but EnableDataIntgPt=0, the XOR removal on reads is not applied, undermining the intended protection or corrupting returned data.

- Command integrity check (CmdIntgCheck) is optional; when off, tampered commands are not detected at this block. |
| 47 | tlul | The M:1 socket routes responses to hosts based purely on the low STIDW bits of d_source provided by the device. It does not validate that d_source corresponds to an in-flight request or the original requester. A malicious or faulty device can misroute responses/errors to a different host by forging d_source low bits. |
| 48 | tlul | The design assumes TL_AIW (IDW) is at least log2(M) (STIDW). If configured with IDW <STIDW, part-selects become invalid/zero-width, undermining identity binding. There is no ASSERT to enforce IDW >= STIDW. |
| 49 | tlul | function get_bad_data_intg returns a vector declared with H2DCmdIntgWidth but computes on DataIntgWidth. Today both widths are 7, so this is harmless; if widths diverge in future, this would mis-size the result and could silently truncate/extend, breaking integrity manipulations that rely on this helper. |
| 50 | tlul | outstanding_txn is a 2-bit counter incremented on a_ack and decremented on d_ack. There is no guard against decrementing from 0. A misbehaving device that emits d_valid without prior a_valid could underflow the counter, potentially prolonging the drain window (StOutstanding) and leading to denial-of-service until reset. Not a confidentiality/integrity bypass, but a robustness gap. |
| 51 | tlul | The host adapter's response data integrity check (EnableRspDataIntgCheck) is parameterized and can be left disabled. In combination with issue #3, if a device misroutes responses, disabled response checking at the host increases the risk of undetected tampering. |

## A.3   ROLES OF AGENTS IN REPORTED ISSUES

An agent may be involved in some capacity for each reported issue. The roles played by agents for each reported issue are shown in Table 4. If the agent was used to determine and localize a security issue, it is tagged as (D,L). If it is used in the flow of identifying the bug but was not the final determinator, it is tagged as Helper (H). If it is used in the flow of identifying a warning, it is tagged as a Warner (W). If it is used in the flow of incorrectly identifying a security issue, it is tagged as False identifier (F). If an agent is not used at all, it is not tagged (-).

Table 4: Roles of Agents in Reported Issues.

| Reported Issue ID | Confirmed & Localized? | CWE | Similar | Assertion | Lint | Anomaly | Simulator |
|---|---|---|---|---|---|---|---|
| 1 | ✓ | H | - | H | D,L | H | - |
| 2 | ✓ | H | - | H | D,L | H | - |
| 3 | ✓ | - | - | - | - | - | D,L |
| 4 | ✓ | - | - | - | - | - | D,L |
| 5 | ⚠ | - | - | - | W | - | - |
| 6 | ✓ | - | - | H | D,L | H | - |
| 7 | ✓ | - | - | D,L | - | - | - |
| 8 | ✓ | H | - | H | D,L | H | - |
| 9 | ✗ | - | - | - | - | - | F |
| 10 | ✗ | F | F | - | - | F | - |
| 11 | ✗ | F | - | - | - | F | - |
| 12 | ✗ | F | - | - | - | F | - |
| 13 | ⚠ | - | - | - | - | W | - |
| 14 | ✗ | - | - | - | - | F | - |
| 15 | ⚠ | W | - | - | - | - | - |
| 16 | ✗ | F | F | - | - | F | - |
| 17 | ⚠ | W | - | - | - | W | - |
| 18 | ⚠ | - | - | - | W | W | W |
| 19 | ⚠ | - | - | - | - | W | - |
| 20 | ✓ | H | - | H | H | H | D,L |
| 21 | ✓ | H | - | H | D,L | H | - |
| 22 | ✓ | H | - | H | D,L | H | - |
| 23 | ✓ | H | - | - | - | H | D,L |
| 24 | ✗ | - | - | - | - | F | - |
| 25 | ⚠ | W | - | - | - | W | - |
| 26 | ✗ | - | - | - | F | - | - |
| 27 | ✗ | - | - | F | - | - | F |
| 28 | ✗ | F | - | - | - | - | - |
| 29 | ⚠ | W | - | - | - | - | - |
| 30 | ✗ | - | - | - | - | F | - |
| 31 | ✓ | - | - | - | - | D,L | - |
| 32 | ✓ | - | - | - | - | D,L | - |
| 33 | ✓ | - | - | - | - | D,L | - |
| 34 | ⚠ | - | - | - | - | W | - |
| 35 | ✗ | - | - | - | - | F | - |
| 36 | ✗ | - | - | - | F | - | - |
| 37 | ✗ | - | - | - | - | F | - |
| 38 | ✓ | H | - | H | D,L | H | - |
| 39 | ✓ | H | - | H | D,L | H | - |
| 40 | ✓ | H | - | H | D,L | H | - |
| 41 | ✓ | - | D,L | - | - | - | - |
| 42 | ✓ | H | - | H | D,L | H | - |
| 43 | ✗ | F | - | - | F | - | - |
| 44 | ⚠ | - | - | - | - | W | - |

| 45 | ✗ | F | - | - | - | - | - |
|----|----|----|----|----|----|----|----|
| 46 | ✗ | F | - | - | - | - | - |
| 47 | ⚠ | W | - | - | - | - | - |
| 48 | ⚠ | W | - | - | - | - | - |
| 49 | ⚠ | - | - | - | - | - | - |
| 50 | ⚠ | - | - | W | W | - | - |
| 51 | ✗ | - | - | - | - | - | - |

### A.4 SECURITY OBJECTIVES AND FILE CATEGORY CLASSES

This appendix describes the classification of security objectives and design files undertaken to investigate the supervisor agent's operational patterns across different runs. Our aim was to determine if recurring tuples of agents and security objectives were present and if their selection followed a logical basis or stochastic distribution. The outcome of this investigation is shown in Figure 7 and analyzed in Section 5. The remainder of this section reports on the assignment methodology for each design file and security objective to their corresponding classes.

#### A.4.1 DESIGN FILE CLASSIFICATION

OpenTitan uses standardized naming for design files where the first part is the IP name, followed by the file type (e.g., `control`, `core`, `reg top`). We classified files based on their postfix:

- **Interface**: `reg_top`, `core_reg_top`, `reg_we_check`, `adapter_reg`, `adapter_sram`, `lci`, `dai`, `kmac_if`, `reg_cdc`, `lc_gate`, `subreg_shadow`.

- **Integration**: `app`, `top`, `core`, `-no_suffix-`.

- **FSM/Control Logic**: `ctrl`, `controller`, `fsm`, `onehot_check`, `sm`, `main_sm`, `cipher_control`.

- **Other**: `part_buf`, `part_unbuf`, `intr`, `intr_hw`, `state_db`, `cmd_stage`, `msgfifo`, `prng_masking`, `ctr_drbg_cmd`, `socket_m1`.

#### A.4.2 SECURITY OBJECTIVE CLASSIFICATION

We collected all security objectives used by the supervisor agent and manually classified them:

- **FSM security**: FSM safety, illegal states, counter rollover, Availability/DoS via stuck states, Find anomalous FSM transitions, state handling, control path: sparse encodings, FSM reset, FSM safe encoding, FSM integrity, FSM illegal state must raise error, FSM error handling, FSM control flow, FSM anomalies, decoded outputs gating across states, FSM robustness, FSM hardening, error handling, unconditional commit

- **access control**: Register access policies, W1C behavior, RO/WO enforcement, Register access policy: ensure write-only CSRs are not readable, address decode matches, no readback of sensitive data, redundancy rails consistency, shadowed register enforcement, Confidentiality: ensure secret key registers are not readable; CWE-200, CWE-668, CWE-126, Confidentiality, integrity: key handling, zeroization, sideload enforcement, CSR policy enforcement, Secret readback prevention: Assert that any read to KEY_SHARE0/1 addresses returns zero, that default read data is zero, RTL security lint: privilege/OTP gating, Check SW register access gates, internal state dump gating, genbits status gating, command ack decoupling from gen path, FIPS flag forcing usage, Map to CWE: privilege escalation (missing authorization), information exposure (status leakage), improper restriction of operations within bounds (DoS), register policy enforcement, register locking, CWE mapping: improper access control, incorrect privilege, write-only secrecy, zeroization gating, no-secret-on-read (keys, zeroize-always-writable, safe-default-read, confidentiality of

secret keys, improper access control, improper error handling
gating zeroization, disable outputs, Fault detection bypass,
shadowed registers, write-one-clear, reserved bit handling,
byte write support, write-ignored timing, shadowed register
update mismatch should trigger recoverable alert, sticky
status bit, check for CWE-1282 improper access control to
FIFO, CWE-1234 missing lock for sensitive operation, map to
CWE: improper access control to secrets, missing privilege
on state read, TOCTOU on REGWEN, debug/TAP isolation, no
unauthorized state transitions; REGWEN/mutex enforcement;
tap isolation in PROD, missing authorization for critical
functions, debug backdoor exposure, token handling, volatile
unlock logic, token comparison consistency, bus access
control, DAI access control, lock enforcement, hidden triggers,
Access control: prove that when read_lock/write_lock
are asserted, DAI cannot issue otp_req_o for disallowed
addresses, Map access-control bypass, hidden trigger to
CWEs, Identify unusual counter/constant triggers that gate
access checks, shadowed register correctness, register
integrity (shadowed write double-commit), shadowed register
double-write integrity must detect mismatches, block commit,
transaction integrity, unintended writes on reset, register
interface misuse, outlier patterns that could break shadowed
register security, life-cycle gating, Life-cycle gating: when
lc_en_i != ON, host sees error responses only; outstanding
txn drain policy, Authorization, identity binding, response
steering integrity, CWE-285/284/345, Address validation/bounds
checks, integrity configuration, CWE-20/285/693, Life-cycle
lock, data confidentiality/integrity when lc disabled,
CWE-284/285/693/1191, outstanding transaction tracking,
life-cycle gating correctness, alert on integrity, register
access control, wipe), data gating, register interface access
control, mutex enforcement, secure wipe, memory/data integrity,
Map the issues to CWEs: disabled secure wipe, W1C/W1S
compliance, CWE mapping, ensure H2D path is blocked

- **entropy**: No entropy leakage to SW without OTP gate, misuse of
entropy configuration

- **masking**: MUBI misuse, incorrect behavior of multi-bit booleans,
MUBI usage, MUBI enable; strict decode on MUBI controls;
alert on invalid MUBI; detect seed drops; health tests
gating, improper input validation (MUBI), mubi misuse

- **side channels**: side-channel unsafe modes, SCA blanking, secure
wipe request/ack sequencing; LSU address blanking must drive
lsu_addr_o when required; ISPR commit must only assert when
instruction executes, SCA blanking bypass, data leakage

- **Other**: reset, latch inference, improper handling of configuration
values, clock domain crossing on control signals, Check
liveness: LP_SLP eventually exits; check cfg values zero
handling; CDC safety of cfg signals, mux default, X-prop,
combinational loops, FIFO hardening, CWE-1245 improper
initialization, CWE-1191 on X-propagation causing control
bypass, insecure reset, race/glitch in control signals,
data loss / DoS conditions, fault signaling, CDC robustness,
metastability, info leakage across reset, glitch resistance,
reset-domain crossing, RTL antipatterns, polarity mistakes,
uninitialized regs, race conditions in arbitration, unusual
patterns

A.5  SEQUENCE OF ACTIONS REPORTS

In this appendix, we provide two reports containing the sequence of actions for ADC ctrl and AES. These logs also contain actions from the executor agents. Figure 5 only considers the supervisor agent actions. All logs with the full outputs of the LLMs are available in our repository.

**Sequence of Actions for ADC ctrl**

```
Listing directory:  <path to>/hackdate/hw/ip/adc_ctrl Listing
directory:  <path to>/hackdate/hw/ip/adc_ctrl/doc Reading file:  <path
to>/hackdate/hw/ip/adc_ctrl/doc/theory_of_operation.md Reading file:  <path
to>/hackdate/hw/ip/adc_ctrl/doc/registers.md Listing directory:  <path
to>/hackdate/hw/ip/adc_ctrl/rtl Reading file with line numbers:  <path
to>/hackdate/hw/ip/adc_ctrl/rtl/adc_ctrl_reg_top.sv Reading file with line
numbers:  <path to>/hackdate/hw/ip/adc_ctrl/rtl/adc_ctrl.sv Reading file with
line numbers:  <path to>/hackdate/hw/ip/adc_ctrl/rtl/adc_ctrl_core.sv Reading
file with line numbers:  <path to>/hackdate/hw/ip/adc_ctrl/rtl/adc_ctrl_fsm.sv
Reading file with line numbers:  <path to>/adc_ctrl/rtl/adc_ctrl_intr.sv
Reading file:  <path to>/hackdate/hw/ip/adc_ctrl/doc/interfaces.md Running
linter agent on <path to>/hackdate/hw/ip/adc_ctrl/rtl/adc_ctrl_fsm.sv for
adc_ctrl_fsm with security objective:  FSM safety, reset, illegal states,
counter rollover, latch inference Running lint checker tool on <path
to>/hackdate/hw/ip/adc_ctrl/rtl/adc_ctrl_fsm.sv for adc_ctrl_fsm with lint
tags:  ['LINT_FSM_WITHOUT_INITIAL_STATE', 'LINT_FSM_UNREACHABLE_STATES',
'LINT_FSM_WITHOUT_EXIT_STATE', 'NoExitFsmState', 'FsmLiveLock', 'NoExitInFSM-ML',
'MissingFsmStateTransition', 'IntReset', 'STARC05-1.3.2.1a', 'STARC05-1.4.3.1b',
'RegisterStuckInResetState-ML', 'FlopFeedbackRace-ML', 'SafeLatch-ML',
'GlitchGenCircuitCheck', 'CounterRollover-ML'] Running llm cwe checker agent
on <path to>/hackdate/hw/ip/adc_ctrl/rtl/adc_ctrl_fsm.sv for adc_ctrl_fsm with
security objective:  Availability/DoS via stuck states, improper handling of
configuration values, and clock domain crossing on control signals Running
llm cwe details retriever tool on Improper or unsafe finite state machine
implementation leading to stuck/locked states or deadlocks (hardware RTL)
Running llm cwe details retriever tool on Improper handling/validation of
configuration register values causing integer underflow/overflow or wraparound
in hardware (e.g., subtracting 1 from 0) Running llm cwe details retriever
tool on Unsynchronized clock domain crossing (CDC) on control/status signals
leading to metastability or data corruption in hardware Running llm cwe
details retriever tool on Unsynchronized clock domain crossing (CDC) leading
to metastability or data corruption in hardware RTL Running assertions
checker agent on <path to>/hackdate/hw/ip/adc_ctrl/rtl/adc_ctrl_fsm.sv for
adc_ctrl_fsm with security objective:  Check liveness:  LP_SLP eventually
exits; check cfg values zero handling; CDC safety of cfg signals Running
assertion checker tool on <path to>/hackdate/hw/ip/adc_ctrl/rtl/adc_ctrl_fsm.sv
for adc_ctrl_fsm with assertions:  <assertions> Running assertion
checker tool on <path to>/hackdate/hw/ip/adc_ctrl/rtl/adc_ctrl_fsm.sv for
adc_ctrl_fsm with assertions:  <assertions> Running verilator agent on
adc_ctrl Running verilator tests for adc_ctrl Running similar bug agent
on <path to>/hackdate/hw/ip/adc_ctrl/rtl/adc_ctrl_fsm.sv for bug:  else if
(wakeup_timer_cnt_q == cfg_wakeup_time_i) begin Running similar bug tool
on <path to>/hackdate/hw/ip/adc_ctrl/rtl/adc_ctrl_fsm.sv for bug:  else
if (wakeup_timer_cnt_q == cfg_wakeup_time_i) begin Reading file with line
numbers:  <path to>/hackdate/hw/ip/adc_ctrl/rtl/adc_ctrl_fsm.sv Running
linter agent on <path to>/hackdate/hw/ip/adc_ctrl/rtl/adc_ctrl_reg_top.sv
for adc_ctrl_reg_top with security objective:  Register access policies,
W1C behavior, RO/WO enforcement, alert on integrity Running lint checker
tool on <path to>/hackdate/hw/ip/adc_ctrl/rtl/adc_ctrl_reg_top.sv
for adc_ctrl_reg_top with lint tags:  ['RegisterInfo-ML', 'W188',
'W415', 'W18', 'DisallowSVAlwaysLatch-ML', 'SafeLatch-ML', 'W502',
'W450L', 'W336L', 'GlitchGenCircuitCheck', 'STARC05-1.2.1.1a',
'LINT_MULTIDRIVEN_BUS', 'W256'] Running anomaly detection agent on <path
to>/hackdate/hw/ip/adc_ctrl/rtl/adc_ctrl_fsm.sv for adc_ctrl_fsm with security
objective:  Find anomalous FSM transitions and state handling Review Agent
invoked
```

**Sequence of Actions for AES**

```
Listing directory:  <path to>hackdate/hw/ip Listing directory:  <path
to>hackdate/hw/ip/aes Listing directory:  <path to>hackdate/hw/ip/aes/doc
Reading file:  <path to>hackdate/hw/ip/aes/doc/theory_of_operation.md Reading
file:  <path to>hackdate/hw/ip/aes/doc/registers.md Listing directory:
<path to>hackdate/hw/ip/aes/rtl Reading file with line numbers:  <path
to>hackdate/hw/ip/aes/rtl/aes_reg_top.sv Running linter agent on <path
to>hackdate/hw/ip/aes/rtl/aes_reg_top.sv for aes_reg_top with security
objective:  Register access policy:  ensure write-only CSRs are not
readable, address decode matches, and no readback of sensitive data Running
lint checker tool on <path to>hackdate/hw/ip/aes/rtl/aes_reg_top.sv for
aes_reg_top with lint tags:  ['CSR_WO_NO_READ', 'CSR_ADDR_DECODE_ONEHOT',
'CSR_SECRET_READBACK', 'CSR_RO_NO_WRITE', 'CSR_SUBWORD_WRITE_PERMIT',
'CSR_READ_MUX_DEFAULT', 'CSR_ADDR_DECODE_MAP', 'CSR_WO_ZERO_READ',
'CSR_REGWEN_GATING', 'CSR_READBACK_MISMATCH'] Running linter agent on <path
```

```
to>hackdate/hw/ip/aes/rtl/aes_control.sv for aes_control with security
objective: FSM hardening and control path: sparse encodings, redundancy
rails consistency, error handling, shadowed register enforcement Running
lint checker tool on <path to>hackdate/hw/ip/aes/rtl/aes_control.sv
for aes_control with lint tags: ['SepFSMLogic', 'FsmUsage-ML',
'STARC05-2.11.3.1', 'UniquePriorityMisuse-ML', 'EnableXPropagation-ML',
'STARC05-2.4.1.4', 'DisallowSVAlwaysLatch-ML', 'W450L', 'STARC05-1.3.2.1a',
'STARC05-1.4.3.1b', 'STARC05-1.2.1.1a', 'W415', 'Rule_3', 'MergeFlops-ML',
'RegisterStuckInResetState-ML', 'HangingFlopOutput-ML', 'OverCasting-ML',
'UseSVCasting-ML', 'LINT_IMPROPER_RANGE_INDEX', 'DetectUnderAndOverFlows',
'TristateSig-ML', 'GlitchGenCircuitCheck', 'TwoStateData-ML'] Running llm
cwe checker agent on <path to>hackdate/hw/ip/aes/rtl/aes_reg_top.sv for
aes_reg_top with security objective: Confidentiality: ensure secret key
registers are not readable; CWE-200, CWE-668, CWE-126 Running llm cwe details
retriever tool on CWE-668 Running llm cwe details retriever tool on CWE-200
Running llm cwe details retriever tool on CWE-126 Reading file with line
numbers: <path to>hackdate/hw/ip/aes/rtl/aes_control.sv Running linter agent
on <path to>hackdate/hw/ip/aes/rtl/aes_core.sv for aes_core with security
objective: Confidentiality and integrity: key handling, zeroization,
sideload enforcement, CSR policy enforcement Running lint checker tool on
<path to>hackdate/hw/ip/aes/rtl/aes_core.sv for aes_core with lint tags:
['UniquePriorityMisuse-ML', 'CheckKeywordsOfCaseStmt-ML', 'SafeLatch-ML',
'DisallowCaseZ-ML', 'SVConstructSanityChecks', 'STARC05-1.4.3.1b',
'STARC05-1.4.1.1', 'STARC05-1.3.2.1a', 'Reset_check11', 'GatedClock',
'DEBUG_LINT_UNUSED_RTL_CODE', 'W215', 'W502', 'STARC05-2.4.1.4'] Running
assertions checker agent on <path to>hackdate/hw/ip/aes/rtl/aes_reg_top.sv
for aes_reg_top with security objective: Secret readback prevention:
Assert that any read to KEY_SHARE0/1 addresses returns zero and
that default read data is zero Running assertion checker tool on
<path to>hackdate/hw/ip/aes/rtl/aes_reg_top.sv for aes_reg_top with
assertions: <assertions> Running assertion checker tool on <path
to>hackdate/hw/ip/aes/rtl/aes_reg_top.sv for aes_reg_top with assertions:
<assertions> Running verilator agent on aes Running verilator tests for aes
Reading file with line numbers: <path to>hackdate/hw/ip/aes/rtl/aes_core.sv
Running similar bug agent on <path to>hackdate/hw/ip/aes/rtl/aes_reg_top.sv
for bug: default: reg_rdata_next = reg2hw.key_share0[0].q; Reading file with
line numbers: <path to>hackdate/hw/ip/aes/rtl/aes_reg_top.sv Running similar
bug tool on <path to>hackdate/hw/ip/aes/rtl/aes_reg_top.sv for bug: default:
reg_rdata_next = reg2hw.key_share0[0].q; Running anomaly detection agent on
<path to>hackdate/hw/ip/aes/rtl/aes_reg_top.sv on aes_reg_top with security
objective: Outliers in register map and data paths that might leak secrets or
violate policies Review Agent invoked
```

A.6 SINGLE-AGENT SYSTEM PROMPT

This section includes the system prompt used for our single-agent analysis and comparison with
MARVEL.

**Supervisor's System Prompt**

```
You are a supervisor agent focused on identifying hardware security
vulnerabilities in RTL code. Your objective is to analyze the given SoC and
generate a detailed, evidence-backed security report.
You have access to the following tools and should use them directly:
DETAILED TOOL INSTRUCTIONS
=== VERILATOR TESTS TOOL ===
Purpose: Execute verilator tests for the given IP and analyze failing test
reports to detect potential security issues.
Usage: run_verilator_tests(ip: str)
Instructions:
- Run a security analysis on the specified IP
- Inspect the logs of failing runs and determine if there are security issues in
the RTL
- If any security issues are found, provide a detailed explanation of the issue
and its location in the RTL code
- Focus on failing test evidence; passing tests don't necessarily rule out
security issues
- Use on IP names after selecting a target

=== ASSERTION CHECKER TOOL ===
Purpose: Execute VC Formal assertions on the top_module to verify security
properties.
Usage: assertion_checker_tool(design_filepath: str, top_module: str,
assertions: dict, clock_signal: str, reset_signal: str, reset_active:
Literal["low", "high"])
Instructions:
- Form relevant SystemVerilog assertions for the RTL under the stated security
objective
- Provide an assertions dictionary name: assertion_string to the tool
- Example structure: "assertion_p1": "property p1;
```

```
n @(posedge clk) signal_A |-> signal_B;
nendproperty
nassertion_p1:  assert property (p1);"
- Include design file, top module, clock/reset signals
- Run the tool and determine if there are security issues from the output
- If there are no falsified assertions, there are no verified security issues
- Cite falsified property locations when found

=== LINT CHECKER TOOL ===
Purpose:  Execute VC SpyGlass Lint checks on the top_module to flag design
violations tied to security concerns.
Usage:  lint_checker_tool(design_filepath:  str, top_module:  str, lint_tags:
List[str])
Instructions:  - First use retrieve_relevant_lint_tags to identify relevant lint
tags for the security objective
- Then run the lint tool on the design file and top module with the identified
tags
- From the tool output, determine if there are security issues
- Treat Error severity as security-impacting
- Map findings to security aspects
- Focus on FSM, uninitialized registers, incorrectly instantiated modules, etc.

=== CWE DETAILS RETRIEVER TOOL ===
Purpose:  Obtain relevant CWE and corresponding details based on the security
issue being analyzed.
Usage:  llm_cwe_details_retriever_tool(security_issue:  str)
Instructions:
- Identify the CWE relevant to the security issue for the given RTL
- Obtain details of the CWE using a concrete security issue description
- Use returned CWE details to guide code inspection
- Then determine if there are security issues relevant to the identified CWE in
the RTL
- Refer to the code that corresponds to the issues identified
- Use to guide deeper analysis and explain impact with CWE context

=== SIMILAR BUG TOOL ===
Purpose:  Look for bugs similar to previously found bugs by searching for similar
code patterns.
Usage:  similar_bug_tool(bug:  str, ip_file:  str)
Instructions:
- Pass a previously found buggy line and a file path to search for similar bugs
- Returns a list of similar bug lines with line numbers
- Treat matches as candidates, not confirmed bugs
- Analyze the identified lines and determine if they are indeed bugs
- Use only after at least one confirmed or strong-signal finding
- Best to use on the same file or equivalent files of different IPs where a bug
was found

=== ANOMALY DETECTOR TOOL ===
Purpose:  Identify anomalous code in RTL through forming clusters of similar
constructs.
Usage:  anomaly_detector_tool(design_filepath:  str)
Instructions:
- Use the anomaly detector tool on the design file to identify anomalous lines
- The tool clusters similar constructs and surfaces outliers that may indicate
unusual or risky patterns
- Treat outliers as leads to review, not definitive bugs
- Determine whether the identified anomalous line(s) represent a security issue
- Confirm with code review or other tools

=== UTILITY TOOLS ===
- list_dir(dir_path):  List directory contents to explore file structure
- read_file(file_path):  Read file content (best for documentation like .md
files)
- read_file_with_line_numbers(file_path):  Read file with line numbers (best for
code files like .sv files)

WORKFLOW GUIDANCE:
Budget and efficiency:
- Start with docs and lint before heavier runs
- Escalate to assertion/verilator after initial signals
- Stop and summarize when budget is exhausted

Constraints and safety:
- Do not use the Similar Bug tool first
- Use it only after at least one confirmed or strong-signal finding
- Treat anomalies/similarities as hypotheses and confirm with code citations or
other tools
ANALYSIS INSTRUCTIONS:
- Read the documentation to identify security features and register interfaces
```

```
policies.
- Use Verilator, Assertion, Anomaly and Linter tools to uncover initial issues in
the design.
- If a bug is detected but not localized, use CWE details to further inspect the
related security aspect in the surrounding RTL.
- After detecting any bugs, use the Similar Bug tool to scan similar files (of
the same or of different IPs) for similar vulnerabilities.

REPORTING FORMAT:
- For each identified issue, provide:
- File name
- Line number(s)
- Brief description of the issue
- Security aspect affected
- Tools used

End your final response with "END".
```

