# OpenReview forum: "MARVEL: Multi-Agent RTL Vulnerability Extraction using Large Language Models"
_ICLR.cc/2026/Conference — Submitted to ICLR 2026_

### Official Review · Reviewer_bfdt · 2025-10-31

**Soundness:** 3
**Presentation:** 3
**Contribution:** 2
**Rating:** 4
**Confidence:** 3

**Summary:**

This paper introduces MARVEL, a multi-agent framework for detecting security vulnerabilities in RTL (Register-Transfer Level) designs. MARVEL follows a Supervisor-Executor architecture, where the supervisor agent orchestrates six executor agents (Linter, Assertion, CWE, Similar-Bug, Anomaly, Simulator), each paired with tools (e.g., Synopsys SpyGlass Lint, VC Formal, Verilator) and small RAG stores. The authors evaluate MARVEL on a buggy dataset from Hack@DATE 2025 using OpenTitan SoC, and MARVEL reports 51 issues, 19 of which are validated as true security vulnerabilities. The authors also provide ablations to compare single-agent vs. supervisor-executor and per-agent contributions.

**Strengths:**

1. The design of MARVEL is clear and intuitive. Agents and toolchains are concretely described, including their prompts, actions, and failure-recovery loops.
2. MARVEL is fully automated and useful for real RTL workflows and complementary to human verification.
3. Evaluation shows that MARVEL can detect security vulnerabilities with a 19/51 TP ratio.

**Weaknesses:**

1. The novelty of this work should be argued more precisely. For example, the paper claims that MARVEL is the "first" multi-agent framework for hardware bug detection. However, SV-LLM is also multi-agent, albeit with weaker agent integration. The authors should clarify the novelty of MARVEL relative to existing studies in terms of what is technically new.
2. The evaluation relies on a non-public SoC, and the full ground-truth bug list is not available. This limits community reproducibility and prevents reporting standard detection metrics (precision/recall/F1).
3. There is no comparison between MARVEL and other recent LLM-for-RTL systems (e.g., SV-LLM, Self-HWDebug) on the same target, leaving MARVEL's relative advantage unclear.

**Questions:**

1. What is the complete ground-truth bug list per IP? Can you report precision/recall/F1?
2. Can you provide a quantitative comparison vs. other recent systems (e.g., SV-LLM) on the same SoC?
3. How does MARVEL perform using only open-source tools, excluding all the proprietary tools?
4. Can you elaborate on the technical novelty of MARVEl compared to other recent studies?

---

> ### Author Response · Authors · 2025-11-20
>
> We thank the reviewer for the valuable feedback.
>
> Commenting on the highlighted weaknesses:
> 1. We expanded the Related Works section to better highlight the novelty of our contribution.
> "... "MARVEL does not focus on explicit vulnerabilities. The security objectives are identified by the supervisor agent from the design documentation. The executor agents receive security objectives from the supervisor and adapt their execution to them using RAG... "
>
> 2. We have compared the buggy Hack@DATE SoC against the closest corresponding commit (by line-level similarity) in the open-source OpenTitan repository. This allowed us to identify all bugs intentionally injected by the competition organizers. Based on this ground truth, we now report the number of true bugs per IP in Table 1 and add precision, recall, and F1-score metrics to Table 2. The updated results are discussed in Sections 4 and 6.
> The overall precision, recall, and F1-score achieved by MARVEL are 0.51, 0.49, and 0.50, respectively. Per-IP results are highly skewed: MARVEL often achieves perfect scores (1-1-1) when actionable issues are present, and 0-0-0 on IPs where hallucinations dominate. This pattern suggests that hallucinations can occasionally lead the framework into unproductive analysis paths, an aspect we highlight as a key direction for future improvement.
>
>
> 3. Only SV-LLM use the same SoC. But their flow is drastically different and does not provide a final security analysis report that we could compare with.   Self-HWDebug focuses on specific CWEs and requires examples of code with and without the vulnerability, meaning that generalization is challenging.
>
> Answering the Questions:
>
> - A1:  We have contacted the competition organizers to obtain the bug list, but they refuse to share the list of inserted bugs. We have compared the buggy Hack@DATE SoC against the closest corresponding commit (by line-level similarity) in the open-source OpenTitan repository. This allowed us to identify all bugs intentionally injected by the competition organizers. Based on this ground truth, we now report the number of true bugs per IP in Table 1 and add precision, recall, and F1-score metrics to Table 2. The updated results are discussed in Sections 4 and 6.
>
> - A2: SV-LLM does not provide a list of identified vulnerabilities, nor any numbers on that matter, making it challenging to compare against it.  Self-HWDebug focuses on specific CWEs and requires examples of code with and without the vulnerability, making generalization difficult.
>
> - A3: We do not have numbers on that, but the Agent is agnostic to the tool that is being called underneath, so the quality of results should only depend on the quality of the tool being called.
>
> - A4. MARVEL produces comprehensive security reports by performing a comprehensive analysis without user intervention. Given the limited scope of previous work and the lack of a clearly identified list of issues, we believe our paper greatly improves on the state of the art.

---

> ### Author Response · Authors · 2025-11-26
>
> We have updated our previous response to reflect the changes included in the latest version of the manuscript.
>
> We believe we have now addressed all concerns raised by the reviewer. We kindly invite further feedback on the revisions, and we remain fully open to making additional improvements if needed. If the reviewer agrees that the changes satisfactorily resolve the concerns, we would appreciate having this reflected in the final evaluation.

---

### Official Review · Reviewer_oM6s · 2025-10-31

**Soundness:** 2
**Presentation:** 3
**Contribution:** 2
**Rating:** 4
**Confidence:** 3

**Summary:**

This paper introduces MARVEL, a multi-agent LLM framework for detecting security vulnerabilities in RTL hardware designs. MARVEL uses a supervisor-executor architecture where a supervisor agent coordinates six specialized executor agents (Linter, Assertion, CWE, Similar Bug, Anomaly, Simulator) that employ different bug detection strategies and integrate with industry EDA tools (VC SpyGlass, VC Formal, Verilator).

**Strengths:**

1. This work claims to be the first comprehensive multi-agent approach for RTL security verification.

2. The paper addresses hardware security verification. The approach handles hardware-specific challenges including clocking, concurrency, hardware CWEs, and FSM properties, and is evaluated on a real-world Hack@DATE benchmark.

3. MARVEL demonstrates solid engineering through practical integration of industry-standard EDA tools (VC SpyGlass Lint, VC Formal, Verilator), iterative refinement for error correction, and logical specialization of six agents combining tool-based and LLM-based approaches.

**Weaknesses:**

1. Table 2 presents security issues classified as 'correct' or 'incorrect,' but the paper never specifies who made these determinations. If the authors themselves judged their system's outputs, this constitutes circular reasoning and lacks objectivity. The absence of inter-rater reliability metrics, independent expert validation, or comparison with the official Hack@DATE answer key fundamentally undermines the credibility of the paper's central claims.

2. The evaluation is fundamentally incomplete without a recall metric. Without knowing the ground truth bug count, it is impossible to assess whether finding 19 bugs represents excellent coverage (e.g., 95% if 20 bugs exist) or poor coverage (e.g., 9.5% if 200 bugs exist). While the authors note that 'ground truth was not disclosed,' they could have requested this information from organizers or conducted independent exhaustive analysis with domain experts. This omission violates standard practices in security research, which require reporting both precision and recall.

3. The paper provides no evidence of generalization beyond a single benchmark. All 12 evaluated IPs come from the OpenTitan family, sharing the same design style, coding conventions, and security policies. Without evaluation on diverse SoCs—such as different RISC-V implementations, commercial IPs, legacy designs, or other application domains—there is significant risk of overfitting to benchmark-specific patterns. This single-benchmark evaluation is insufficient to support the paper's claims of general applicability.

4. The paper's efficiency claims lack quantitative support due to the absence of a human expert baseline. The authors claim MARVEL 'saves hours' compared to manual analysis, but provide no empirical evidence. With 37% precision, experts must manually validate 32 false positives—potentially requiring 160 minutes at 5 minutes per issue, costing approximately `$400` in expert time plus `$3` in LLM costs. Without demonstrating that this total cost/time is lower than direct expert analysis, the efficiency claims remain speculative.

5. The paper completely lacks statistical significance testing. Despite GPT-5's inherent stochasticity (temperature=0.15), each IP was evaluated only once, with no reporting of variance, confidence intervals, or results from multiple runs with different random seeds. The reported results could represent lucky outliers rather than typical performance.

6. The paper lacks essential comparisons with state-of-the-art methods. FLAG, a publicly available baseline, was not evaluated on the same benchmark. Additionally, the paper provides no comparison with: (1) pure tool usage (VC SpyGlass/VC Formal without LLM assistance), (2) single-agent LLM with tool access, or (3) other claimed baselines.

7. The ablation study is incomplete and potentially biased. It evaluates only three IPs—the same ones used for model selection in Section 3.2—creating risk of data leakage. The study is missing systematic evaluations of: supervisor-only configurations, optimal 2-agent and 3-agent combinations, and all-agents-except-worst configurations. Given that the Similar Bug Agent contributes minimally (1 D&L) and the Anomaly Agent primarily generates false positives (16 FI vs. 3 D&L), it remains unclear whether all six agents are necessary.

8. The supervisor's decision-making process remains opaque. The paper states that the supervisor may call agents in any order and multiple times' but provides no specification of selection criteria, validation of decision quality, or analysis of optimality. Figure 5 shows that approximately 35% of actions involve navigation (List Dir, Read File), suggesting potential inefficiency. The non-deterministic workflow raises reproducibility concerns, and while Figure 7 displays call frequency, it does not assess decision quality.

9. The analysis of false positives is inadequate. While the authors note that 'many false positives arose from security suggestions rather than concrete flaws,' this explanation is insufficient. The paper lacks a systematic taxonomy of the 32 false positives by type, root cause, or responsible agent. Notably, the Anomaly Agent produced 16 false identifications (84% FP rate) and the CWE Agent produced 14, but no explanation is provided for these high rates. Without pattern identification across file types, security objectives, or agent combinations, iterative improvement is hindered.

10. The formulation of security objectives is underspecified and unvalidated. The paper claims that 148 security properties were 'correctly formulated,' but the formulation process remains opaque—is it automatic, does it involve manual review, or was expert validation performed? Without comparison to expert-defined ground truth, it is impossible to assess completeness (are critical properties missing?) or relevance (are irrelevant properties included?). Poor property formulation directly leads to searching for incorrect bugs or missing important vulnerabilities

11. The anomaly detection methodology is questionable. The approach uses DBSCAN clustering to identify outliers as potential bugs, but outliers do not necessarily represent bugs—unique but valid design patterns may be incorrectly flagged. The selection of hyperparameters (eps, min_samples) is unjustified, and semantic embeddings may miss security context (e.g., similar syntactic structures with different security implications). The empirical results confirm this weakness: the Anomaly Agent achieves an 84% false positive rate (16 FI vs. 3 D&L). Better alternatives exist, including AST-based pattern matching, data flow analysis, and supervised learning approaches.

12. The paper provides insufficient analysis of the multi-agent architecture's benefits. The comparison with a single-agent baseline is limited to three IPs and lacks depth. Missing analyses include: the value of redundancy when multiple agents identify the same bug, complementarity in detecting different bug types, coordination overhead (approximately 35% spent on navigation), diminishing returns as agents are added, conflict resolution mechanisms, and optimal agent ordering. Without this analysis, it is impossible to determine whether the multi-agent architcture's complexity is justified compared to simpler alternatives, such as a single agent with multiple tools or a reduced agent set.

**Questions:**

- What is the core methodological contribution beyond adapting existing multi-agent frameworks (Lee et al., 2024) to a new domain? The supervisor-executor architecture, RAG-based retrieval, embeddings-based similarity, DBSCAN clustering, and tool-calling APIs are all standard techniques. Could the authors clarify what algorithmic innovations, theoretical insights, or generalizable methodological advances distinguish this from an engineering application of existing methods?

- Without novel ML methods or strong empirical validation (currently limited by single-benchmark evaluation, missing baselines, and lack of statistical testing), how does this work meet the research contribution standards of a top-tier ML venue versus being more appropriate for hardware security conferences or industry tracks?

---

> ### Author Response · Authors · 2025-11-20
>
> We thank the reviewer for providing valuable feedback.
>
> 1. (This also addresses point 9) We reclassified MARVEL's findings in three categories: Bugs, Warnings and Hallucinations. We now clearly define these categories and explain how we analyzed the results.
>  "We used content analysis sessions, similar to prior work in
> software engineering (Catolino et al., 2019), to perform this classification. Two authors of this
> work independently reviewed each bug report and the relevant design files. This includes RTL files,
> documentation, and test logs. When necessary, the OpenTitan repository (lowRISC contributors,
> 2023) was used as a golden reference. Then, a discussion was held to resolve any differences and
> reach a consensus. "
>
> 2. We have compared the buggy Hack@DATE SoC against the closest corresponding commit (by line-level similarity) in the open-source OpenTitan repository. This allowed us to identify all bugs intentionally injected by the competition organizers. Based on this ground truth, we now report the number of true bugs per IP in Table 1 and add precision, recall, and F1-score metrics to Table 2. The updated results are discussed in Sections 4 and 6.
>
> 3. At present, OpenTitan is one of the few publicly available SoCs with a realistic, manually curated set of security bugs, which limits the breadth of empirical evaluation. Importantly, MARVEL does not use any OpenTitan-specific heuristics or tuning: the agents operate on generic prompts, hardware CWEs, tool outputs, and design documentation. We therefore expect MARVEL to transfer to designs with different architectures and coding styles. We have clarified this in the manuscript and identified broader multi-SoC evaluation as a key direction for future work.
>
> 4. We added some comments in the results section to address this concern.
> "The findings are well distributed through all the IPs and
> the number of reported issues and hallucinations is relatively small, highlighting that MARVEL
> is effective at filtering the noisy outputs of the base tools. Assuming an engineering effort of
> 20 minutes per finding, the most expensive IP analysis would take 140 minutes, which is a fraction of
> the man-months allocated to security verification. Moreover, integrating this analysis
> during the design phase would reduce the number of findings per run."
>
> 5. Due to the computational cost of running MARVEL end-to-end across all IPs, we were unable to perform full multi-seed evaluations within the review timeline. However, MARVEL’s agents rely heavily on tool-grounded signals (lint, simulation logs, documentation parsing), which substantially reduces the impact of model stochasticity, especially at temperature 0.15. We acknowledge that multi-seed evaluations and confidence intervals would strengthen the statistical rigor of the results, and we have added a clarification in the manuscript identifying this as a key direction for future work.
>
> 6. State-of-the-art comparison: We have expanded Section 5 (Related Works) to clearly differentiate MARVEL from prior work. Unlike FLAG and Self-HWDebug, which target predefined vulnerabilities, MARVEL autonomously identifies security objectives from documentation and adapts verification strategies via RAG. Direct comparison is not feasible due to: (1) different problem scopes (specific CWEs vs. general security verification), (2) unavailable benchmarks/detailed results from prior work, and (3) lack of public implementations.
> Tool-only comparison: Section 4.2 provides ablation studies showing single-executor runs (effectively tool-only) yield lower true positive ratios than MARVEL (Figures 9-10). The supervisor's cross-validation across tools is key to filtering false positives.
> Single-agent comparison: We do provide this in Section 4.2, Figure 8. MARVEL achieves equal or better bug identification than single-agent GPT-5 with identical tool access, with more nuanced outputs (warnings vs. hallucinations) aiding triage.
> Human+tools comparison: While valuable, this requires human study infrastructure beyond rebuttal scope. However, MARVEL completes comprehensive IP analysis in 18-53 minutes versus typical person-months for manual verification.
> Unlike prior work, we provide complete transparency: all 51 issues with classifications (Table 3), agent contributions (Table 4), and open-source implementation.
>
> 7. The three IPs span diverse functionality (I/O, crypto, memory) and bug densities (2, 9, 10 bugs), providing representative coverage. We added an ablation study with all but one executors.
> Our analysis reveals that agents with fewer direct determinations still provide critical support: Similar Bug serves as a helper in pattern propagation after initial detection, while Anomaly acts as a "hypothesis generator" triggering deeper investigation by other agents. Figure 9 shows that excluding any single agent degrades performance, though we acknowledge the optimal subset remains an open question.

---

> ### Author Response · Authors · 2025-11-26
>
> We have updated our previous response to reflect the changes included in the latest version of the manuscript, and continue our response here due to character constraints:
>
> 8. The supervisor's decision-making is guided by its system prompt (Figure 3), which provides heuristics for tool selection while allowing LLM-based planning autonomy. Appendix A.5 provides complete action sequences showing the workflow: explore documentation to extract security objectives, then systematically call executors. Figure 7 demonstrates logical decisions—high-frequency pairings like "access control + interface files" align with security engineering practice, while nonsensical combinations never occur.
> The 35% navigation overhead (Figure 5) is appropriate rather than inefficient—these lightweight operations (completing in seconds vs. minutes for verification tools) enable understanding design structure and extracting security policies before verification decisions, mirroring human engineering practice.
> We acknowledge non-determinism from LLM sampling (temperature=0.15) raises reproducibility concerns. However, the supervisor consistently: (1) extracts correct objectives, (2) calls each executor at least once, and (3) completes within predictable bounds (18-53 min).
>
> 9.  Addressed with 1.
>
> 10. The supervisor agent automatically extracts security objectives from design documentation (Section 2.2)—it reads specification files (e.g., theory_of_operation.md, registers.md) and derives relevant security properties. Our statement that objectives were "correctly formulated" comes from manual validation by two paper authors who compared extracted objectives against OpenTitan documentation during the bug classification process (Section 3.2), verifying they were: (1) accurately derived, (2) relevant to the module, and (3) technically sound. We added this validation methodology explicitly to Section 4.1.
>
> 11. The Anomaly Agent is designed as a hypothesis generator rather than a definitive bug detector. Its role is to surface unusual patterns for review by the supervisor and other agents—not to make final determinations. This explains why it frequently serves in Helper (H) and Warner (W) roles (Figure 6) rather than as a sole determination. Despite high standalone false positives, the multi-agent architecture allows the supervisor to filter anomaly-generated hypotheses through other verification methods (linting, assertions, CWE analysis). Figure 9 shows that excluding the Anomaly Agent degrades overall performance, suggesting it provides value within the ensemble even with noisy outputs. We agree that AST-based pattern matching, data flow analysis, or supervised methods could improve precision. However, these require significant domain-specific engineering or labeled training data (unavailable for hardware security). The embedding-based approach provides a zero-shot, design-agnostic baseline that integrates easily with our LLM framework.
>
> 12. In MARVEL, redundancy is leveraged by the supervisor to confirm the presence of bugs while conflicting information is used to perform deeper analysis. While we do not have a formal conflict-resolution mechanism, the benefit of the supervisor agent is its flexibility and ability to adapt to context to make the best decision in a given scenario.
> Many of the techniques leveraged by the executor agents (simulation, linting, assertion) are industry standard approaches that present distinct trade-offs. For example, linting is efficient and can effectively find common issues such as missing state transitions, but can be noisy due to high false positives. On the other hand, Simulation is costly but can find complex semantic errors and generally provides concrete results. By combining these two, we can replicate the comprehensive verification approach used in the industry. While the scale of MARVEL increases its complexity, our analysis shows that this is beneficial. As discussed in the paper, we do not explicitly state an optimal agent order to the supervisor. This is because the "optimal" analysis steps are driven by the security requirements identified and the results found by executor agents. We found that the supervisor agent typically executed the most appropriate next action in most cases. Evaluating this would require a user study based on feedback from experienced security verification engineers, which we leave for future work.

---

> ### Author Response · Authors · 2025-11-26
>
> ```What is the core methodological contribution beyond adapting existing multi-agent frameworks (Lee et al., 2024) to a new domain? The supervisor-executor architecture, RAG-based retrieval, embeddings-based similarity, DBSCAN clustering, and tool-calling APIs are all standard techniques. Could the authors clarify what algorithmic innovations, theoretical insights, or generalizable methodological advances distinguish this from an engineering application of existing methods?```
>
> We acknowledge that MARVEL builds on established multi-agent patterns and does not introduce novel algorithmic techniques. Our contribution is in the systematic integration and domain adaptation required for hardware security verification, which presents unique challenges absent in software domains:
>
> Hardware-specific challenges: Unlike software bugs addressed by Lee et al. (2024), RTL vulnerabilities involve hardware semantics (clocking, concurrency, state machines), require specialized EDA tools (VC Formal, SpyGlass, Verilator), and must reason about security properties embedded in hardware specifications. Section 1 notes: "RTL bugs are deeply tied to hardware semantics...which make them harder to detect and repair with traditional software-centric approaches."
>
> Domain-specific executor design: Each executor agent embodies domain expertise: the Linter Agent maps security objectives to 1255 hardware-specific lint rules via RAG (Section 2.3), the Assertion Agent generates SystemVerilog temporal properties with canonical structures for formal verification (Section 2.4), and the CWE Agent retrieves hardware-relevant CWEs from MITRE's database (Section 2.5). These are not generic tool wrappers but carefully designed verification strategies.
>
> Unified verification workflow: MARVEL demonstrates the first end-to-end framework integrating diverse hardware verification methodologies (formal, linting, simulation, static analysis) under autonomous LLM control. Prior work uses LLMs with individual tools but lacks unified orchestration. Section 1 states this addresses "a gap in autonomous thinking...by using LLMs in an agentic workflow."

---

> ### Author Response · Authors · 2025-11-26
>
> ```Without novel ML methods or strong empirical validation (currently limited by single-benchmark evaluation, missing baselines, and lack of statistical testing), how does this work meet the research contribution standards of a top-tier ML venue versus being more appropriate for hardware security conferences or industry tracks?```
>
> We appreciate this fundamental question. We submitted to the Applications to Physical Sciences track specifically because MARVEL bridges ML methods and hardware security, a critical domain in physical systems. MARVEL addresses core challenges in agentic AI systems: multi-agent coordination for complex technical tasks, LLM-based planning and tool use, handling noisy tool outputs through ensemble reasoning, and long-context information synthesis. Demonstrating how agentic LLM frameworks can automate specialized technical verification in physical systems, with both capabilities (50% F1) and clear limitations (35% hallucination rate, decision-making opacity), provides valuable insights for ML applications in safety-critical domains beyond software, which is precisely the scope of this track.

---

> ### Author Response · Authors · 2025-11-26
>
> We believe we have now addressed all concerns raised by the reviewer. We kindly invite further feedback on the revisions, and we remain fully open to making additional improvements if needed. If the reviewer agrees that the changes satisfactorily resolve the concerns, we would appreciate having this reflected in the final evaluation.

---

### Official Review · Reviewer_qdLp · 2025-11-03

**Soundness:** 2
**Presentation:** 2
**Contribution:** 2
**Rating:** 4
**Confidence:** 5

**Summary:**

Hardware security verification is complex and time-consuming, relying on formal verification, linting, and simulation tests. To enhance this process, the authors propose MARVEL, a multi-agent LLM framework that unifies reasoning, decision-making, and tool use. MARVEL imitates how a designer identifies security flaws in RTL code through a supervisor agent that defines security policies and delegates tasks to executor agents. These agents use formal tools, linters, simulations, LLM-based detection, and static analysis to find bugs and report results for validation. Tested on an OpenTitan-based SoC, MARVEL detected 51 issues, with 19 confirmed security vulnerabilities.

**Strengths:**

(1) This idea is straightforward and can be easily understood.

(2) The experiments explore multiple GPT models for detecting hardware vulnerabilities, which is interesting.

**Weaknesses:**

(1) This paper focuses on the use of Large Language Models (LLMs) for hardware security; however, it remains unclear how the authors specifically leverage these models for vulnerability detection. The methodology section lacks sufficient detail on how the LLMs are integrated into the detection pipeline, what kind of data or prompts are used, and how the outputs are analyzed or validated.

(2) My another concern about this paper  is the novelty issue. The paper appears to directly apply existing LLMs (e.g., GPT-5) to the hardware security domain without providing new insights or deeper understanding. Moreover, there are many existing works that have explored similar topics using LLMs for hardware vulnerability detection. The paper fails to discuss how it relates to or differs from these prior studies, which makes its contribution seem quite limited.

(3) The experimental evaluation is also insufficient. For example, the paper only investigates GPT-based models, but does not consider other popular alternatives such as Meta’s LLaMA 3 or Google’s Gemini, which could provide a more comprehensive comparison for vulnerability detection.

(4) Finally, the writing quality requires significant improvement. In several critical sections, it is difficult to understand what the authors actually did in terms of experimentation and analysis, or what motivated their methodological choices. Clarifying these aspects would strengthen the paper considerably.

**Questions:**

Please refer to my comments for more details.

---

> ### Author Response · Authors · 2025-11-20
>
> We thank the reviewer for providing valuable feedback.
>
> (1) We addressed this point by rewriting how we present the Agents in Section 2. We now highlight the main novelty and contributions of each agent, then explain how they are achieved through implementation details. The original manuscript includes examples of execution flows in Appendix A1; these should provide the reader with concrete steps taken by each agent to achieve its task.
>
> (2) We have expanded the Related Works section, better highlighting the improvements that our paper makes on the state of the art. Specifically, "MARVEL does not focus on explicit vulnerabilities. The security objectives are identified by the supervisor agent from the design documentation. The executor agents receive security objectives from the supervisor and adapt their execution to them using RAG."
>
> (3) We added experiments with Gemini 2.5 Pro in our model selection. We used the same settings and system prompts. While we noted differences in the report's writing style, we found results comparable to those of GPT 4.1. We believe this now better supports our claim that the framework is usable with different models.
>
> (4) We have added new experiments and improved Sections 4 and 5, explaining better how we carried out the classification and methodological choices. We improved the results discussions and readability. We now include standard metrics such as precision, recall, and F1 score as suggested by other reviewers.

---

> ### Author Response · Authors · 2025-11-26
>
> We have updated our previous response to reflect the changes included in the latest version of the manuscript.
>
> We believe we have now addressed all concerns raised by the reviewer. We kindly invite further feedback on the revisions, and we remain fully open to making additional improvements if needed. If the reviewer agrees that the changes satisfactorily resolve the concerns, we would appreciate having this reflected in the final evaluation.

---

### Author Response · Authors · 2025-11-20
**Manuscript update**

We thank the reviewers for the detailed feedback and suggestions.

We have revised and improved our manuscript to address your comments and concerns.
We are still working to add an analysis running MARVEL using Gemini 2.5 pro. We will upload another revised manuscript early next week, in the meanwhile we kindly ask for feedback on the current revisions:


- We reclassified MARVEL's findings in three categories: Bugs, Warnings and Hallucinations. Multiple reviewers and external industry experts raised concerns about our previously binary classification. We believe this shows a more detailed and accurate picture of the quality of results found by MARVEL.
- We updated the results section accordingly and expanded our discussion of the results in section 4. We also improved the flow and organization to make it more digestible.
- We restructured how we present the agents in section 2. Highlighting their main features, how we implement them and the tools used.
- We expanded the related work section to better discuss the current state of the art and how MARVEL improves upon it.

---

### Author Response · Authors · 2025-11-24
**Bug List Concern**

Dear reviewers,

We have compared the buggy Hack@DATE SoC against the closest corresponding commit (by line-level similarity) in the open-source OpenTitan repository. This allowed us to identify all bugs intentionally injected by the competition organizers. Based on this ground truth, we now report the number of true bugs per IP in Table 1 and add precision, recall, and F1-score metrics to Table 2. The updated results are discussed in Sections 4 and 6.

The overall precision, recall, and F1-score achieved by MARVEL are 0.51, 0.49, and 0.50, respectively. Per-IP results are highly skewed: MARVEL often achieves perfect scores (1-1-1) when actionable issues are present, and 0-0-0 on IPs where hallucinations dominate. This pattern suggests that hallucinations can occasionally lead the framework into unproductive analysis paths, an aspect we highlight as a key direction for future improvement.

---

### Author Response · Authors · 2025-11-26
**Updated Manuscript**

We thank the reviewers for their thorough and constructive feedback. We have revised the manuscript accordingly and believe that the updated version substantially improves the quality and clarity of the work.

Below, we summarize the main changes. We will also provide point-by-point responses to each reviewer to address all specific comments.

- We refined MARVEL’s output taxonomy into three categories Bugs, Warnings, and Hallucinations, in response to concerns from reviewers and industry experts about the previous binary scheme. This provides a more precise and informative characterization of MARVEL’s findings.
- We added the number of known bugs present in each benchmark IP to Table 1.
- We included precision, recall, and F1 metrics in Table 2.
- We extended the model selection study to include results for Gemini 2.5 Pro.
- We added an all-but-one executor study to broaden the ablation analysis of executor impact.
- We updated the results section and expanded the discussion in Section 4, improving flow, readability, and interpretability.
- We reorganized Section 2 to more clearly present the agents, detailing their core functionalities, implementation, and tool usage.
- We expanded the related work section to better situate MARVEL within the state of the art and articulate its contributions.

Tables 1 and 2 and Figures 4, 6, and 8 have been updated. The previous Figure 9 is now Figure 10, and a new Figure 9 has been added.

We believe these revisions substantially strengthen the manuscript and address the concerns raised. We appreciate the reviewers’ careful evaluation and hope the improvements are reflected in their final assessment.

---

### Meta-Review · Area_Chair_wccv · 2026-01-09

**Summary:**

The reviewers agree that the paper presents a well-engineered multi-agent framework (MARVEL) for RTL hardware security verification, integrating LLM-based agents with industry-standard EDA tools. The system is clearly motivated, addresses a practically important problem, and demonstrates non-trivial engineering effort through its supervisor–executor architecture and evaluation on the Hack@DATE OpenTitan benchmark. Some reviewers acknowledge that the framework can identify real vulnerabilities and complement human verification workflows.

However, the primary concerns center on limited research novelty and insufficient empirical rigor for a top-tier ML venue. Multiple reviewers question whether MARVEL introduces methodological contributions beyond adapting existing multi-agent LLM frameworks and standard techniques (e.g., tool calling, RAG, clustering) to a new application domain. While the authors clarify in the rebuttal that the contribution lies in systematic integration and domain adaptation, reviewers remain divided on whether this constitutes a sufficiently strong research contribution rather than an application-oriented system paper.

The evaluation methodology is another major point of contention. Reviewers raised concerns about reliance on a single benchmark (OpenTitan), lack of diversity in evaluated designs, incomplete baselines, and initially missing standard metrics such as recall and statistical significance testing. The authors addressed several of these issues in the rebuttal by adding precision/recall/F1 metrics, additional experiments, and clearer validation procedures. Nonetheless, concerns remain regarding generalization beyond OpenTitan, limited comparisons to competing systems, and the absence of statistical robustness analysis.

Finally, reviewers expressed reservations about efficiency claims, reproducibility, and decision transparency in the multi-agent system. Although the rebuttal provides clarifications on supervisor behavior, agent roles, and validation procedures, aspects such as cost–time trade-offs, non-determinism, and justification of architectural complexity remain only partially resolved.

Overall, the paper is viewed as a solid and practically motivated system with meaningful engineering contributions, but reviewers disagree on whether its methodological novelty, empirical breadth, and analytical depth are sufficient to meet the acceptance bar of a top-tier ML conference.

**Reviewer Concerns:**

Several reviewer concerns were partially or substantially addressed through the authors’ responses and revisions. In particular, the authors clarified the overall system design and agent roles, expanded the related work discussion, and improved the clarity of the methodology and experimental sections. They added standard evaluation metrics (precision, recall, F1), reported additional experiments , and provided more detailed explanations of how security objectives are derived, validated, and processed by the supervisor and executor agents. These revisions improve the paper’s readability and address earlier ambiguity regarding validation procedures.

However, several substantive concerns remain unresolved. Most notably, questions regarding research novelty persist. While the authors explicitly acknowledge that MARVEL does not introduce new algorithmic techniques and instead emphasizes system integration and domain adaptation, some reviewers remain unconvinced that this constitutes a sufficiently strong methodological contribution for a top-tier ML venue, as opposed to an application- or systems-oriented contribution.

Concerns about empirical scope and generalization also remain. The evaluation is still largely confined to the OpenTitan/Hack@DATE benchmark, with limited evidence that the framework generalizes to other SoCs, design styles, or hardware security contexts. Comparisons to alternative LLM-for-RTL systems remain incomplete, and the lack of statistical significance analysis or multi-run evaluation continues to limit confidence in the robustness of the reported results.

Finally, reviewers continue to express reservations about system transparency, efficiency, and reproducibility. Although the rebuttal clarifies the supervisor’s heuristics and agent interactions, aspects such as non-determinism, cost–time trade-offs relative to human experts, and justification of the multi-agent architecture’s complexity versus simpler baselines are not fully quantified. These unresolved issues collectively temper confidence in the practical and scientific impact of the proposed framework.

**Reviewer Scores:**

Reviewer qdLp (Score: 4)

This reviewer raised concerns regarding unclear methodology, limited novelty, insufficient evaluation, and writing quality. The authors’ revisions improve clarity, expand related work, and add additional experiments and metrics, which partially address these concerns. However, the core issues regarding novelty and empirical breadth remain. The score might remain the same or improve slightly, but a substantial upward change appears unlikely.

Reviewer oM6s (Score: 4)

This reviewer expressed deep concerns about evaluation rigor, objectivity, statistical validity, and justification of the multi-agent architecture. While the rebuttal clarifies several design choices and adds metrics, many fundamental issues—such as single-benchmark evaluation, limited baselines, lack of statistical testing, and incomplete justification of system complexity—are only partially resolved. It is therefore unlikely that the reviewer’s score would materially change.

Reviewer bfdt (Score: 4)

This reviewer viewed the system as clear and practically useful but questioned the strength of the novelty claims and the lack of comprehensive comparisons. The authors’ responses help clarify positioning and improve evaluation reporting, which may strengthen confidence in the engineering contribution. Nevertheless, concerns about limited generalization and comparative evaluation likely persist. The score would most likely remain unchanged.

---

### Decision · Program_Chairs · 2026-01-26

Reject